# Reconciling functional differences in populations of neurons recorded with two-photon imaging and electrophysiology

Joshua H Siegle[1†], Peter Ledochowitsch[1†], Xiaoxuan Jia[1], Daniel J Millman[1], Gabriel K Ocker[1‡], Shiella Caldejon[1], Linzy Casal[1], Andy Cho[1], Daniel J Denman[2§], Séverine Durand[1], Peter A Groblewski[1], Gregg Heller[1], India Kato[1], Sara Kivikas[1], Jérôme Lecoq[1], Chelsea Nayan[1], Kiet Ngo[2], Philip R Nicovich[2#], Kat North[1], Tamina K Ramirez[1], Jackie Swapp[1], Xana Waughman[1], Ali Williford[1], Shawn R Olsen[1], Christof Koch[1], Michael A Buice[1], Saskia EJ de Vries[1]*

[1]MindScope Program, Allen Institute, Seattle, United States; [2]Allen Institute for Brain Science, Allen Institute, Seattle, United States

*For correspondence:
saskiad@alleninstitute.org

[†]These authors contributed equally to this work

Present address: [‡]Boston University, Boston, United States; [§]University of Colorado Denver Anschutz Medical Campus, Aurora, United States; [#]Cajal Neuroscience, Seattle, United States

Competing interests: The authors declare that no competing interests exist.

**Abstract** Extracellular electrophysiology and two-photon calcium imaging are widely used methods for measuring physiological activity with single-cell resolution across large populations of cortical neurons. While each of these two modalities has distinct advantages and disadvantages, neither provides complete, unbiased information about the underlying neural population. Here, we compare evoked responses in visual cortex recorded in awake mice under highly standardized conditions using either imaging of genetically expressed GCaMP6f or electrophysiology with silicon probes. Across all stimulus conditions tested, we observe a larger fraction of responsive neurons in electrophysiology and higher stimulus selectivity in calcium imaging, which was partially reconciled by applying a spikes-to-calcium forward model to the electrophysiology data. However, the forward model could only reconcile differences in responsiveness when restricted to neurons with low contamination and an event rate above a minimum threshold. This work established how the biases of these two modalities impact functional metrics that are fundamental for characterizing sensory-evoked responses.

## Introduction

Systems neuroscience aims to explain how complex adaptive behaviors can arise from the interactions of many individual neurons. As a result, population recordings—which capture the activity of multiple neurons simultaneously—have become the foundational method for progress in this domain. Extracellular electrophysiology and calcium-dependent two-photon optical physiology are by far the most prevalent population recording techniques, due to their single-neuron resolution, ease of use, and scalability. Recent advances have made it possible to record simultaneously from thousands of neurons with electrophysiology (*Jun et al., 2017*; *Siegle et al., 2021*; *Stringer et al., 2019a*) or tens of thousands of neurons with calcium imaging (*Sofroniew et al., 2016*; *Stringer et al., 2019b*; *Weisenburger et al., 2019*). While insights gained from both methods have been invaluable to the field, it is clear that neither technique provides a completely faithful picture of the underlying neural activity. In this study, our goal is to better understand the inherent biases of each recording modality, and specifically how to appropriately compare results obtained with one method to those obtained with the other.

Head-to-head comparisons of electrophysiology and imaging data are rare in the literature, but are critically important as the practical aspects of each method affect their suitability for different experimental questions. Since the expression of calcium indicators can be restricted to genetically defined cell types, imaging can easily target recordings to specific sub-populations (*Madisen et al., 2015*). Similarly, the use of retro- or anterograde viral transfections to drive indicator expression allows imaging to target sub-populations defined by their projection patterns (*Glickfeld et al., 2013*; *Gradinaru et al., 2010*). The ability to identify genetically or projection-defined cell populations in electrophysiology experiments is far more limited (*Economo et al., 2018*; *Jia et al., 2019*; *Lima et al., 2009*). Both techniques have been adapted for chronic recordings, but imaging offers the ability to reliably return to the same neurons over many days without the need to implant bulky hardware (*Peters et al., 2014*). Furthermore, because imaging captures structural, in addition to functional, data, individual neurons can be precisely registered to tissue volumes from electron microscopy (*Bock et al., 2011*; *Lee et al., 2016*), in vitro brain slices (*Ko et al., 2011*), and potentially other ex vivo techniques such as in situ RNA profiling (*Chen et al., 2015*). In contrast, the sources of extracellular spike waveforms are very difficult to localize with sufficient precision to enable direct cross-modal registration.

Inherent differences in the spatial sampling properties of electrophysiology and imaging are widely recognized, and influence what information can be gained from each method (*Figure 1A*). Multi-photon imaging typically yields data in a single plane tangential to the cortical surface, and is limited to depths of <1 mm due to a combination of light scattering and absorption in tissue. While multi-plane (*Yang et al., 2016*) and deep structure (*Ouzounov et al., 2017*) imaging are both areas of active research, imaging of most subcortical structures requires physical destruction of more superficial tissues (*Dombeck et al., 2010*; *Feinberg and Meister, 2015*; *Skocek et al., 2018*). Extracellular electrophysiology, on the other hand, utilizes microelectrodes embedded in the tissue, and thus dense recordings are easiest to perform along a straight line, normal to the cortical surface, in order to minimize per-channel tissue displacement. Linear probes provide simultaneous access to neurons in both cortex and subcortical structures, but make it difficult to sample many neurons from the same cortical layer.

The temporal resolutions of these two methodologies also differ in critical ways (*Figure 1B*). Imaging is limited by the dwell time required to capture enough photons to distinguish physiological changes in fluorescence from noise (*Svoboda and Yasuda, 2006*), and the kinetics of calcium-dependent indicators additionally constrain the ability to temporally localize neural activity (*Chen et al., 2013*). While kilohertz-scale imaging has been achieved (*Kazemipour et al., 2019*; *Zhang et al., 2019*), most studies are based on data sampled at frame rates between 1 and 30 Hz. In contrast, extracellular electrophysiology requires sampling rates of 20 kHz or higher, in order to capture the action potential waveform shape that is essential for accurate spike sorting. High sampling rates allow extracellular electrophysiology to pin-point neural activity in time with sub-millisecond resolution, enabling analyses of fine-timescale synchronization across simultaneously recorded neural populations. The fact that electrophysiology can measure action potentials—what we believe to be the fundamental currency of neuronal communication and causation—bestows upon it a more basic ontological status than on calcium imaging, which captures an indirect measure of a neuron's spike train.

To date, there has been no comprehensive attempt to characterize how the choice of recording modality affects the inferred functional properties of neurons in sensory cortex. Our limited understanding of how scientific conclusions may be skewed by the recording modality represents the weakest link in the chain of information integration across the techniques available to neurophysiologists today. To address this, we took advantage of two recently collected large-scale datasets that sampled neural activity in mouse visual cortex using either two-photon calcium imaging (*de Vries et al., 2020*) or dense extracellular electrophysiology (*Siegle et al., 2021*). These datasets were collected using standardized pipelines, such that the surgical methods, experimental steps, and physical geometry of the recording rigs were matched as closely as possible (*Figure 1C,D*). The overall similarity of these Allen Brain Observatory pipelines eliminates many of the potential confounding factors that arise when comparing results from imaging and electrophysiology experiments. We note that this is not an attempt at calibration against ground truth data, but rather an attempt to reconcile results across two uniquely comprehensive datasets collected under highly standardized conditions.

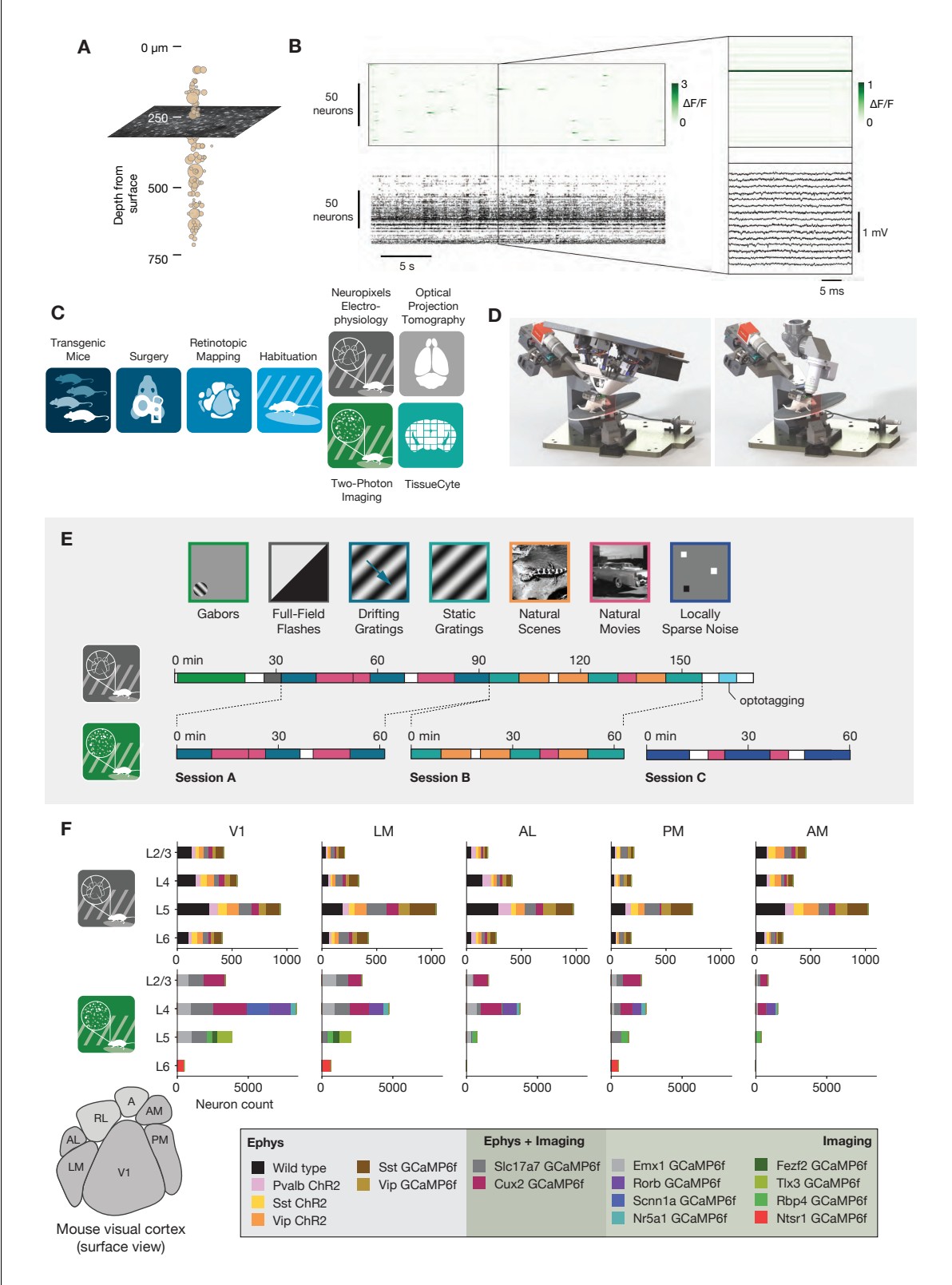

**Figure 1.** Overview of the ephys and imaging datasets. (**A**) Illustration of the orthogonal spatial sampling profiles of the two modalities. Black and white area represents a typical imaging plane (at approximately 250 μm below the brain surface), while tan circles represent the inferred locations of cortical neurons recorded with a Neuropixels probe (area is proportional to overall amplitude). (**B**) Comparison of temporal dynamics between the modalities. *Top*: a heatmap of ΔF/F values for 100 neurons simultaneously imaged in V1 during the presentation of a 30 s movie clip. *Bottom*: raster plot for 100

*Figure 1 continued on next page*

*Figure 1 continued*

neurons simultaneously recorded with a Neuropixels probe in V1 in a different mouse viewing the same movie. *Inset:* Close-up of one sample of the imaging heatmap, plotted on the same timescale as 990 samples from 15 electrodes recorded during the equivalent interval from the ephys experiment. (C) Steps in the two data generation pipelines. Following habituation, mice proceed to either two-photon imaging or Neuropixels electrophysiology. (D) Side-by-side comparison of the rigs used for ephys (left) and imaging (right). (E) Schematic of the stimulus set used for both modalities. The ephys stimuli are shown continuously for a single session, while the imaging stimuli are shown over the course of three separate sessions. (F) Histogram of neurons recorded in each area and layer, grouped by mouse genotype.

Our comparison focused on metrics that capture three fundamental features of neural responses to environmental stimuli: (1) responsiveness, (2) preference (i.e. the stimulus condition that maximizes the peak response), and (3) selectivity (i.e. sharpness of tuning). Responsiveness metrics characterize whether or not a particular stimulus type (e.g. drifting gratings) reproducibly elicits increased activity. For responsive neurons, preference metrics (e.g. preferred temporal frequency) determine which stimulus condition (out of a finite set) elicits the *largest* response, and serve as an indicator of a neuron's functional specialization—for example, whether it responds preferentially to slow- or fast-moving stimuli. Lastly, selectivity metrics (e.g. orientation selectivity, lifetime sparseness) characterize a neuron's ability to distinguish between particular exemplars within a stimulus class. All three of these features must be measured accurately in order to understand how stimuli are represented by individual neurons.

We find that preference metrics are largely invariant across modalities. However, in this dataset, electrophysiology suggests that neurons show a higher degree of responsiveness, while imaging suggests that responsive neurons show a higher degree of selectivity. In the absence of steps taken to mitigate these differences, the two modalities will yield mutually incompatible conclusions about basic neural response properties. These differences could be reduced by lowering the amplitude threshold for valid ΔF/F events, applying a spikes-to-calcium forward model to the electrophysiology data (*Deneux et al., 2016*), or sub-selection of neurons based either on event rate or by contamination level (the likelihood that signal from other neurons is misattributed to the neurons under consideration). This reconciliation reveals the respective biases of these two recording modalities, namely that extracellular electrophysiology predominantly captures the activity of highly active units while missing or merging low-firing-rate units, while calcium-indicator binding dynamics sparsify neural responses and supralinearly amplify spike bursts.

## Results

We compared the visual responses measured in the Allen Brain Observatory Visual Coding ('imaging') and Allen Brain Observatory Neuropixels ('ephys') datasets, publicly available through brain-map.org and the AllenSDK Python package. These datasets consist of recordings from neurons in six cortical visual areas (as well as subcortical areas in the Neuropixels dataset) in the awake, head-fixed mouse in response to a battery of passively viewed visual stimuli. For both datasets, the same drifting gratings, static gratings, natural scenes, and natural movie stimuli were shown (*Figure 1E*). These stimuli were presented in a single 3 hr recording session for the ephys dataset. For the imaging dataset, these stimuli were divided across three separate 1 hr imaging sessions from the same group of neurons. In both ephys and imaging experiments, mice were free to run on a rotating disc, the motion of which was continuously recorded.

The imaging dataset was collected using genetically encoded GCaMP6f (*Chen et al., 2013*) under the control of specific Cre driver lines. These Cre drivers limit the calcium indicator expression to specific neuronal populations, including different excitatory and inhibitory populations found in specific cortical layers (see *de Vries et al., 2020* for details). The ephys dataset also made use of transgenic mice in addition to wild-type mice. These transgenic mice expressed either channelrhodopsin in specific inhibitory populations for identification using optotagging (see *Siegle et al., 2021* for details), or GCaMP6f in specific excitatory or inhibitory populations (see Materials and methods). Unlike in the imaging dataset, however, these transgenic tools did not determine which neurons could be recorded.

We limited our comparative analysis to putative excitatory neurons from five cortical visual areas (V1, LM, AL, PM, and AM). In the case of the imaging data, we only included data from 10 excitatory

Cre lines, while for ephys we limited our analysis to regular-spiking units by setting a threshold on the waveform duration (>0.4 ms). After this filtering step, we were left with 41,578 neurons from 170 mice in imaging, and 11,030 neurons from 52 mice in ephys. The total number of cells for each genotype, layer, and area is shown in *Figure 1F*.

## Calculating response magnitudes for both modalities

In order to directly compare results from ephys and imaging, we first calculated the magnitude of each neuron's response to individual trials, which were defined as the interval over which a stimulus was present on the screen. We computed a variety of metrics based on these response magnitudes, and compared the overall distributions of those metrics for all the neurons in each visual area. The methods for measuring these responses necessarily differ between modalities, as explained below.

For the ephys dataset, stimulus-evoked responses were computed using the spike times identified by Kilosort2 (*Pachitariu et al., 2016a*; *Stringer et al., 2019a*). Kilosort2 uses information in the extracellularly recorded voltage traces to find templates that fit the spike waveform shapes of all the units in the dataset, and assigns a template to each spike. The process of 'spike sorting'—regardless of the underlying algorithm—does not perfectly recover the true underlying spike times, and has the potential to miss spikes (false negatives) or assign spikes (or noise waveforms) to the wrong unit (false positives). The magnitude of the response for a given trial was determined by counting the total number of spikes (including false positives and excluding false negatives) that occured during the stimulation interval. This spike-rate–based analysis is the de facto standard for analyzing electrophysiology data, but it washes out information about bursting or other within-trial dynamics. For example, a trial that includes a four-spike burst will have the same apparent magnitude as a trial with four isolated spikes (*Figure 2A*).

Methods for determining response magnitudes for neurons in imaging datasets are less standardized, and deserve careful consideration. The most commonly used approach involves averaging the continuous, baseline-normalized fluorescence signal over the trial interval. This method relies on information that is closer to the raw data. However, it suffers the severe drawback that, due to the long decay time of calcium indicators, activity from one trial can contaminate the fluorescence trace during the next trial, especially when relatively short (<1 s) inter-stimulus intervals are used. To surmount this problem, one can attempt to determine the onset of abrupt changes in fluorescence and analyze these extracted 'events,' rather than the continuous trace. There are a variety of algorithms available for this purpose, including non-negative deconvolution (*Vogelstein et al., 2010*; *Friedrich et al., 2017*), approaches that model calcium binding kinetics (*Deneux et al., 2016*; *Greenberg et al., 2018*), and methods based on machine learning (*Theis et al., 2016*; *Berens et al., 2018*; *Rupprecht et al., 2021*). For our initial comparison, we extracted events using the same method we applied to our previous analysis of the large-scale imaging dataset (*de Vries et al., 2020*). This algorithm finds event times by reframing $\ell_0$-regularized deconvolution as a change point detection problem that has a mathematically guaranteed, globally optimal 'exact' solution (hereafter, 'exact $\ell_0$'; *Jewell and Witten, 2018*; *Jewell et al., 2018*). The algorithm includes a sparsity constraint (λ) that is calibrated to each neuron's overall noise level.

For the most part, the events that are detected from the 2P imaging data do not represent individual spikes, but rather are heavily biased towards indicating short bouts of high firing rate, for example bursting (*Huang et al., 2021*). There is, however, rich information contained in the amplitudes of these events, which have a non-linear—albeit on average monotonic—relationship with the underlying number of true spikes within a window. Therefore, in our population imaging dataset, we calculated the trial response magnitude by summing the amplitudes of events that occurred during the stimulation interval (*Figure 2B*). In example trials for the same hypothetical neuron recorded with both modalities (*Figure 2A,B*), the response magnitudes are equivalent from the perspective of electrophysiology. However, from the perspective of imaging, the trial that includes a spike burst (which results in a large influx of calcium) may have an order-of-magnitude larger response than a trial that only includes isolated spikes.

## Baseline metric comparison

A comparison between individual neurons highlights the effect of differences in response magnitude calculation on visual physiology. A spike raster from a neuron in V1 recorded with electrophysiology

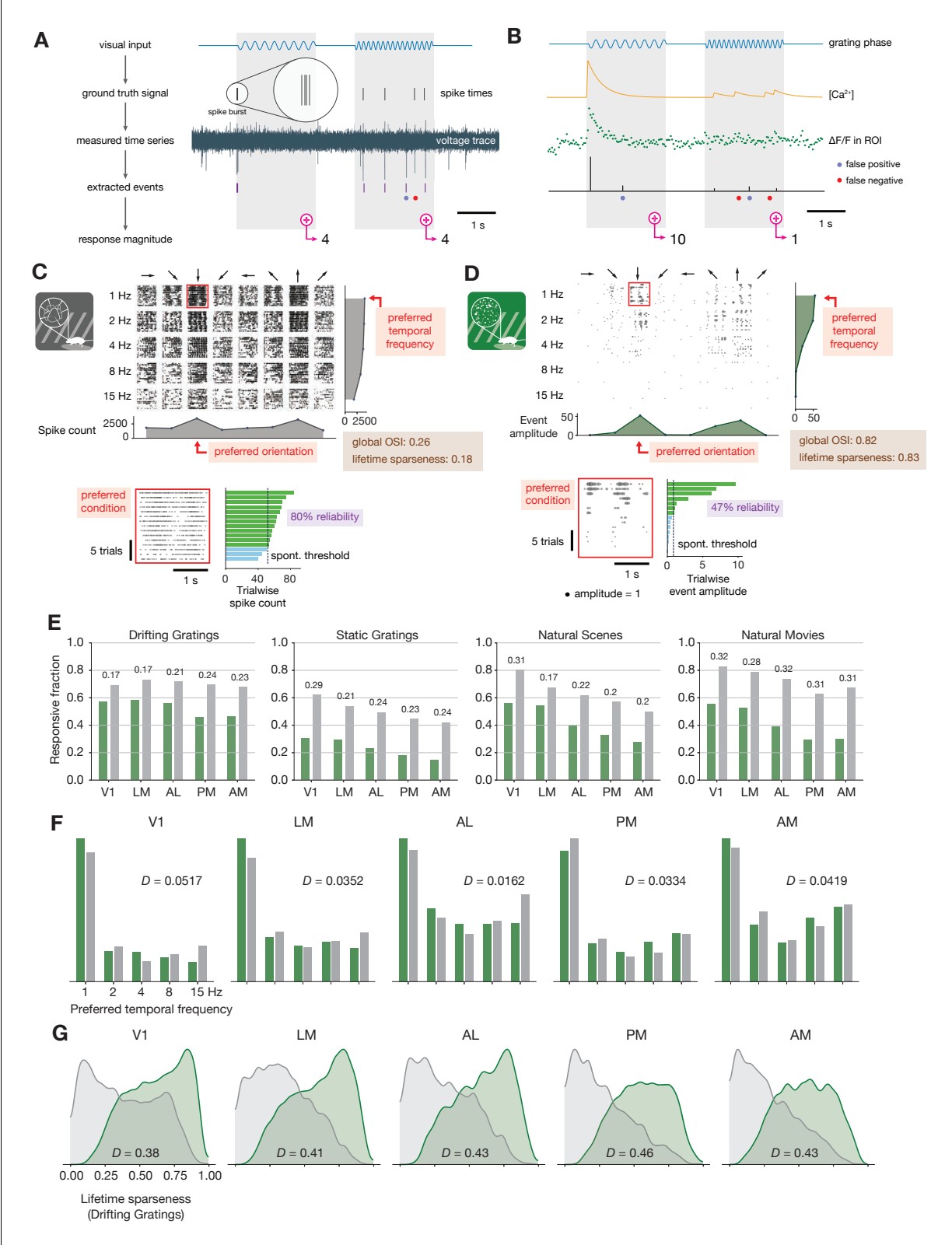

**Figure 2.** Baseline metric comparison. (**A**) Steps involved in computing response magnitudes for units in the ephys dataset. (**B**) Same as A, but for the imaging dataset. (**C**) Drifting gratings spike rasters for an example ephys neuron. Each raster represents 2 s of spikes in response to 15 presentations of a drifting grating at one orientation and temporal frequency. Inset: spike raster for the neuron's preferred condition, with each trial's response magnitude shown on the right, and compared to the 95th percentile of the spontaneous distribution. Responsiveness (purple), preference (red), and

*Figure 2 continued on next page*

*Figure 2 continued*

selectivity (brown) metrics are indicated. (**D**) Same as C, but for an example imaged neuron. (**E**) Fraction of neurons deemed responsive to each of four stimulus types, using the same responsiveness metric for both ephys (gray) and imaging (green). Numbers above each pair of bars represent the Jensen–Shannon distance between the full distribution of response reliabilities for each stimulus/area combination. (**F**) Distribution of preferred temporal frequencies for all neurons in five different areas. The value *D* represents the Jensen–Shannon distance between the ephys and imaging distributions. (**G**) Distributions of lifetime sparseness in response to a drifting grating stimulus for all neurons in five different areas. The value *D* represents the Jensen–Shannon distance between the ephys and imaging distributions.

The online version of this article includes the following figure supplement(s) for figure 2:

**Figure supplement 1.** Baseline comparisons for additional preference and selectivity metrics.
**Figure supplement 2.** Matching laminar distribution patterns across modalities.
**Figure supplement 3.** Characterizing the impact of running behavior on response metrics.
**Figure supplement 4.** Matching running behavior across modalities.

(*Figure 2C*) appears much denser than the corresponding event raster for a separate neuron that was imaged in the same area (*Figure 2D*). For each neuron, we computed responsiveness, preference, and selectivity metrics. We consider both neurons to be responsive to the drifting gratings stimulus class because they have a significant response (p < 0.05, compared to a distribution of activity taken during the epoch of spontaneous activity) on at least 25% of the trials of the preferred condition (the grating direction and temporal frequency that elicited the largest mean response) (*de Vries et al., 2020*). Since these neurons were deemed responsive according to this criterion, their function was further characterized in terms of their preferred stimulus condition and their selectivity (a measure of tuning curve sharpness). We use lifetime sparseness (*Vinje and Gallant, 2000*) as our primary selectivity metric, because it is a general metric that is applicable to every stimulus type. It reflects the distribution of responses of a neuron across some stimulus space (e.g. natural scenes or drifting gratings), equaling 0 if the neuron responds equivalently to all stimulus conditions, and one if the neuron only responds to a single condition. Across all areas and mouse lines, lifetime sparseness is highly correlated with more traditional selectivity metrics, such as drifting gratings orientation selectivity ($R = 0.8$ for ephys, 0.79 for imaging; Pearson correlation), static gratings orientation selectivity ($R = 0.79$ for ephys, 0.69 for imaging), and natural scenes image selectivity ($R = 0.85$ for ephys, 0.95 for imaging).

For our initial analysis, we sought to compare the results from ephys and imaging *as they are typically analyzed in the literature*, prior to any attempt at reconciliation. We will refer to these comparisons as 'baseline comparisons' in order to distinguish them from subsequent comparisons made after applying one or more transformations to the imaging and/or ephys datasets. We pooled responsiveness, preference, and selectivity metrics for all of the neurons in a given visual area across experiments, and quantified the disparity between the imaging and ephys distributions using Jensen–Shannon distance. This is the square root of the Jensen–Shannon divergence, which is a method of measuring the disparity between two probability distributions that is symmetric and always has a finite value (*Lin, 1991*). Jensen–Shannon distance is equal to 0 for perfectly overlapping distributions, and one for completely non-overlapping distributions, and falls in between these values for partially overlapping distributions.

Across all areas and stimuli, the fraction of responsive neurons was higher in the ephys dataset than the imaging dataset (*Figure 2E*). To quantify the difference between modalities, we computed the Jensen–Shannon distance for the distributions of response reliabilities, rather than the fraction of responsive neurons at the 25% threshold level. This is done to ensure that our results are not too sensitive to the specific responsiveness threshold we have chosen. We found tuning preferences to be consistent between the two modalities, including preferred temporal frequency (*Figure 2F*), preferred direction (*Figure 2—figure supplement 1A*), preferred orientation (*Figure 2—figure supplement 1B*), and preferred spatial frequency (*Figure 2—figure supplement 1C*). This was based on the qualitative similarity of their overall distributions, as well as their low values of Jensen–Shannon distance. Selectivity metrics, such as lifetime sparseness (*Figure 2G*), orientation selectivity (*Figure 2—figure supplement 1D*), and direction selectivity (*Figure 2—figure supplement 1E*), were consistently higher in imaging than ephys.

## Controlling for laminar sampling bias and running behavior

To control for potential high-level variations across the imaging and ephys experimental preparations, we first examined the effect of laminar sampling bias. For example, the ephys dataset contained more neurons in layer 5, due to the presence of large, highly active cells in this layer. The imaging dataset, on the other hand, had more neurons in layer 4 due to the preponderance of layer 4 Cre lines included in the dataset (*Figure 2—figure supplement 2A*). After resampling each dataset to match layer distributions (*Figure 2—figure supplement 2B*, see Materials and methods for details), we saw very little change in the overall distributions of responsiveness, preference, and selectivity metrics (*Figure 2—figure supplement 2C–E*), indicating that laminar sampling biases are likely not a key cause of the differences we observed between the modalities.

We next sought to quantify the influence of behavioral differences on our comparison. As running and other motor behavior can influence visually evoked responses (*Niell and Stryker, 2010*; *Stringer et al., 2019a*; *Vinck et al., 2015*; *de Vries et al., 2020*), could modality-specific behavioral differences contribute to the discrepancies in the response metrics? In our datasets, mice tend to spend a larger fraction of time running in the ephys experiments, perhaps because of the longer experiment duration, which may be further confounded by genotype-specific differences in running behavior (*Figure 2—figure supplement 3A*). Within each modality, running had a similar impact on visual response metrics. On average, units in ephys and neurons in imaging have slightly lower responsiveness during periods of running versus non-running (*Figure 2—figure supplement 3B*), but slightly higher selectivity (*Figure 2—figure supplement 3C*). To control for the effect of running, we sub-sampled our imaging experiments in order to match the overall distribution of running fraction to the ephys data (*Figure 2—figure supplement 4A*). This transformation had a negligible impact on responsiveness, selectivity, and preference metrics (*Figure 2—figure supplement 4B–D*). From this analysis we conclude that, at least for the datasets examined here, behavioral differences do not account for the differences in functional properties inferred from imaging and ephys.

## Impact of event detection on functional metrics

We sought to determine whether our approach to extracting events from the 2P data could explain between-modality differences in responsiveness and selectivity. Prior work has shown that scientific conclusions can depend on both the method of event extraction and the chosen parameters (*Evans et al., 2019*). However, the impact of different algorithms on functional metrics has yet to be assessed in a systematic way. To address this shortcoming, we first compared two event detection algorithms, exact $\ell_0$ and unpenalized non-negative deconvolution (NND), another event extraction method that performs well on a ground truth dataset of simultaneous two-photon imaging and loose patch recordings from primary visual cortex (*Huang et al., 2021*). Care was taken to ensure that the characteristics of the ground truth imaging data matched those of our large-scale population recordings in terms of their imaging resolution, frame rate, and noise levels, which implicitly accounted for differences in laser power across experiments.

The correlation between the ground truth firing rate and the overall event amplitude within a given time bin is a common way of assessing event extraction performance (*Theis et al., 2016*; *Berens et al., 2018*; *Rupprecht et al., 2021*). Both algorithms performed equally well in terms of their ability to predict the instantaneous firing rate (for 100 ms bins, exact $\ell_0$ $r = 0.48 \pm 0.23$; NND $r = 0.50 \pm 0.24$; $p = 0.1$, Wilcoxon signed-rank test). However, this metric does not capture all of the relevant features of the event time series. In particular, it ignores the rate of false positive events that are detected in the absence of a true underlying spike (*Figure 3A*). We found that the exact $\ell_0$ method, which includes a built-in sparsity constraint, had a low rate of false positives ($8 \pm 2\%$ in 100 ms bins, $N = 32$ ground truth recordings), whereas NND had a much higher rate ($21 \pm 4\%$; $p = 8e-7$, Wilcoxon signed-rank test). Small-amplitude false positive events have very little impact on the overall correlation between the ground truth spike rate and the extracted events, so parameter optimization does not typically penalize such events. However, we reasoned that the summation of many false positive events could have a noticeable impact on response magnitudes averaged over trials. Because these events always have a positive sign, they cannot be canceled out by low-amplitude negative deflections of similar magnitude, as would occur when analyzing $\Delta F/F$ directly.

We tested the impact of applying a minimum-amplitude threshold to the event time series obtained via NND. If the cumulative event amplitude within a given time bin (100 ms) did not exceed

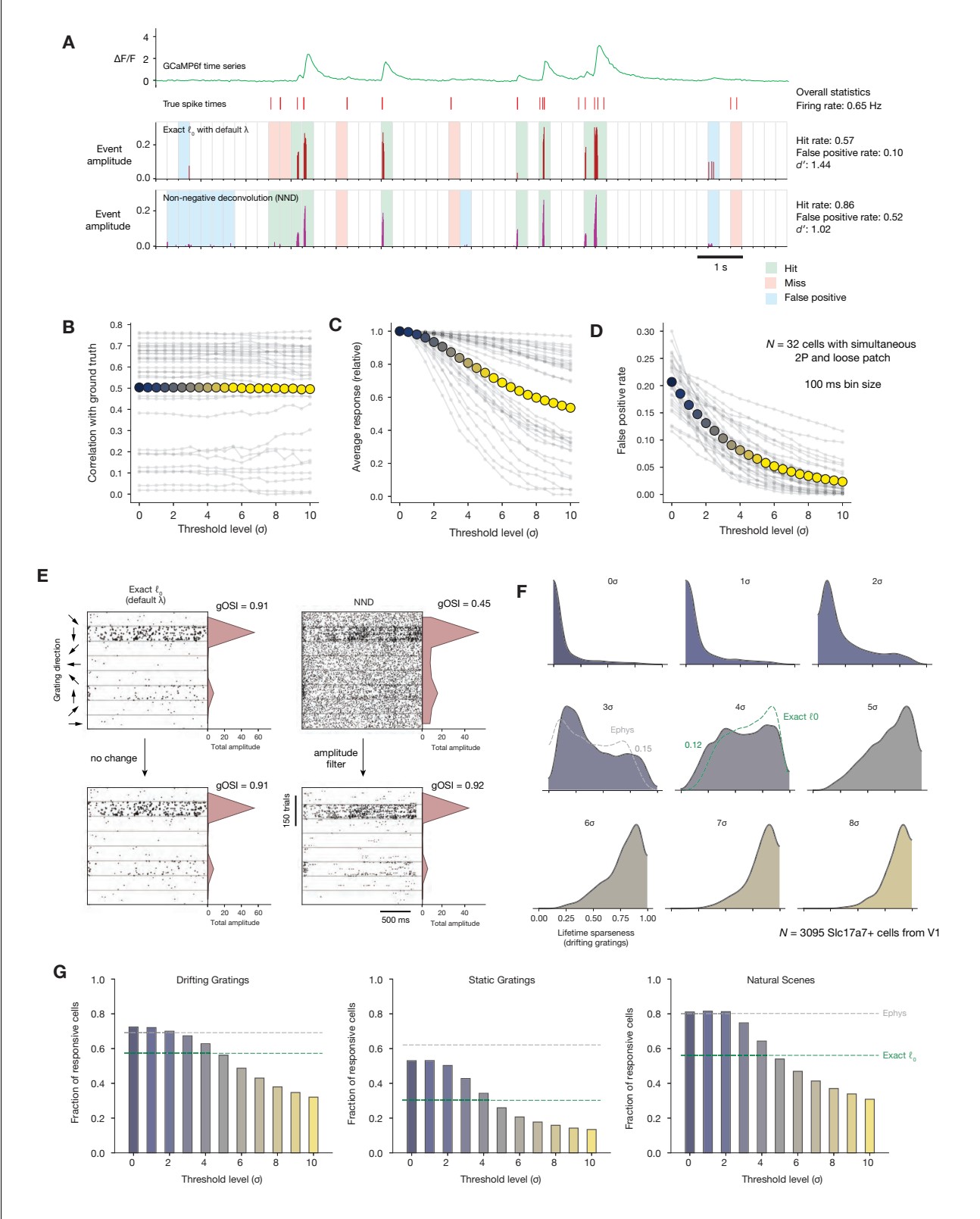

**Figure 3.** Impact of calcium event detection on functional metrics. (**A**) GCaMP6f fluorescence trace, spike times, and detected events for one example neuron with simultaneous imaging and electrophysiology. Events are either detected via exact $\ell_0$-regularized deconvolution (*Jewell and Witten, 2018*) or unpenalized non-negative deconvolution (*Friedrich et al., 2017*). Each 250 ms bin is classified as a 'hit,' 'miss,' 'false positive,' or 'true positive' by comparing the presence/absence of detected events with the true underlying spike times. (**B**) Correlation between binned event amplitude and ground

*Figure 3 continued on next page*

*Figure 3 continued*

truth firing rate after filtering NND events at different threshold levels relative to each neuron's noise level (σ). The average across neurons is shown as colored dots. (C) Same as B, but for the average response amplitude within a 100 ms interval (relative to the amplitude with no thresholding applied). (D) Same as B, but for the probability of detecting a false positive event in a 100 ms interval. (E) Event times in response to 600 drifting gratings presentations for one example neuron, sorted by grating direction (ignoring differences in temporal frequency). Dots representing individual events are scaled by the amplitude of the associated event. Each column represents the results of a different event detection method. The resulting tuning curves and measured global orientation selectivity index (gOSI) are shown to the right of each plot. (F) Lifetime sparseness distributions for Scl17a7+ V1 neurons, after filtering out events at varying threshold levels relative to each neuron's baseline noise level. The overall V1 distributions from ephys (gray) and exact $\ell_0$ (green) are overlaid on the most similar distribution. (G) Fraction of Slc17a7+ V1 neurons responding to three different stimulus types, after filtering out events at varying threshold levels relative to each neuron's baseline noise level.

a threshold (set to a multiple of each neuron's estimated noise level), the events in that window were removed. As expected, this procedure resulted in almost no change in the correlation between event amplitude and the ground truth firing rate (*Figure 3B*). However, it had a noticeable impact on both the average response magnitude within a given time window (*Figure 3C*), as well as the false positive rate (*Figure 3D*).

Applying the same thresholding procedure to an example neuron from our population imaging dataset demonstrates how low-amplitude events can impact a cell's apparent selectivity level. The prevalence of such events differs between the two compared approaches to event extraction, the exact $\ell_0$ method used in *Figure 2* and NND with no regularization (*Figure 3E*). When summing event amplitudes over many drifting gratings presentations, the difference in background rate has a big impact on the measured value of global orientation selectivity (gOSI), starting from 0.91 when using the exact $\ell_0$ and dropping to 0.45 when using NND. However, these differences could be reconciled simply by setting a threshold to filter out low-amplitude events.

Extracting events from a population dataset ($N$ = 3095 Slc17a7+ neurons from V1) using NND resulted in much lower measured overall selectivity levels, even lower than for electrophysiology (*Figure 3F*). Thresholding out events at multiples of each neuron's noise level (σ) raised selectivity levels; a threshold between 3 and 4σ brought the selectivity distribution closest to the ephys distribution, while a threshold between 4 and 5σ resulted in selectivity that roughly matched that obtained with exact $\ell_0$ (*Figure 2G*). The rate of low-amplitude events also affected responsiveness metrics (*Figure 3G*). Responsiveness was highest when all detected events were included, matching or slightly exceeding the levels measured with ephys. Again, applying an amplitude threshold between 4 and 5σ brought responsiveness to the level originally measured with exact $\ell_0$. By imposing the same noise-based threshold on the minimum event size that we originally computed to determine optimal regularization strength (essentially performing a post-hoc regularization), we are able to reconcile the results obtained with NND with those obtained via exact $\ell_0$.

This analysis demonstrates that altering the 2P event extraction methodology represents one possible avenue for reconciling results from imaging and ephys. However, as different parameters were needed to reconcile either selectivity or responsiveness, and the optimal parameters further depend on the presented stimulus class, this cannot be the whole story. Furthermore, relying only on 2P event extraction parameters to reconcile results across modalities implies that the ephys data is itself unbiased, and all we need to do is adjust our imaging analysis pipeline until our metrics match. Because we know this is not the case, we explored the potential impacts of additional factors on the discrepancies between our ephys and imaging datasets.

## Controlling for transgene expression

Given that imaging (but not ephys) approaches fundamentally require the expression of exogenous proteins (e.g. Cre, tTA, and GCaMP6f in the case of our transgenic mice) in specific populations of neurons, we sought to determine whether such foreign transgenes, expressed at relatively high levels, could alter the underlying physiology of the neural population. All three proteins have been shown to have neurotoxic effects under certain conditions (*Han et al., 2012*; *Schmidt-Supprian and Rajewsky, 2007*; *Steinmetz et al., 2017*), and calcium indicators, which by design bind intracellular calcium, can additionally interfere with cellular signaling pathways. To examine whether the expression of these genes could explain the differences in functional properties inferred from imaging and ephys experiments, we performed electrophysiology in mice that expressed GCaMP6f under the

control of specific Cre drivers. We collected data from mice with GCaMP6f expressed in dense excitatory lines (Cux2 and Slc17a7) or in sparse inhibitory lines (Vip and Sst), and compared the results to those obtained from wild-type mice (*Figure 4A*). On average, we recorded 45.9 ± 7.5 neurons per area in 17 wild-type mice, and 55.8 ± 15.6 neurons per area in 19 GCaMP6f transgenic mice (*Figure 4B*). The distribution of firing rates of recorded neurons in mice from all Cre lines was similar to the distribution for units in wild-type mice (*Figure 4C*).

Because some GCaMP mouse lines have been known to exhibit aberrant seizure-like activity (*Steinmetz et al., 2017*), we wanted to check whether spike bursts were more prevalent in these mice. We detected bursting activity using the LogISI method, which identifies bursts in a spike train based on an adaptive inter-spike interval threshold (*Pasquale et al., 2010*). The dense excitatory Cre lines showed a slight increase in burst fraction (the fraction of all spikes that participate in bursts) compared to wild-type mice (*Figure 4D*). This minor increase in burstiness, however, was not associated with changes in responsiveness or selectivity metrics that could account for the baseline differences between the ephys and imaging datasets. The fraction of responsive neurons was not lower in the GCaMP6f mice, as it was for the imaging dataset—in fact, in some visual areas there was an increase in responsiveness in the GCaMP6f mice compared to wild-type (*Figure 4E*). In addition, the distribution of selectivities was largely unchanged between wild-type and GCaMP6f mice (*Figure 4F*). Thus, while there may be subtle differences in the underlying physiology of GCaMP6f mice, particularly in the dense excitatory lines, those differences cannot explain the large discrepancies in visual response metrics derived from ephys or imaging.

## Forward-modeling synthetic imaging data from experimental ephys data

Given the substantial differences between the properties of extracellularly recorded spikes and events extracted from fluorescence traces (*Figure 2A–C*), and the potential impact of event extraction parameters on derived functional metrics (*Figure 3*), we hypothesized that transforming spike trains into simulated calcium events could reconcile some of the baseline differences in response metrics we have observed. The inverse transformation—converting fluorescence events into synthetic spike times—is highly under-specified, due to the reduced temporal resolution of calcium imaging (*Figure 1B*).

To implement the spikes-to-calcium transformation, we used MLSpike, a biophysically inspired forward model (*Deneux et al., 2016*). MLSpike explicitly considers the cooperative binding between GCaMP and calcium to generate synthetic ΔF/F fluorescence traces using the spike trains for each unit recorded with ephys as input. We extracted events from these traces using the same exact $\ell_0$-regularized detection algorithm applied to our experimental imaging data, and used these events as inputs to our functional metrics calculations (*Figure 5A*).

A subset of the free parameters in the MLSpike model (e.g. ΔF/F rise time, Hill parameter, saturation parameter, and normalized resting calcium concentration) were fit to simultaneously acquired loose patch and two-photon-imaging recordings from layer 2/3 of mouse visual cortex (*Huang et al., 2021*). Additionally, three parameters were calibrated on the fluorescence traces from the imaging dataset to capture the neuron-to-neuron variance of these parameters: the average amplitude of a fluorescence transient in response to a spike burst (*A*), the decay time of the fluorescence transients (*τ*), and the level of Gaussian noise in the signal (*σ*) (*Figure 5B*). For our initial characterization, we selected parameter values based on the mode of the overall distribution from the imaging dataset.

The primary consequence of the forward model was to 'sparsify' each neuron's response by washing out single spikes while non-linearly boosting the amplitude of 'bursty' spike sequences with short inter-spike intervals. When responses were calculated on the ephys spike train, a trial containing a 4-spike burst within a 250 ms window would have the same magnitude as a trial with four isolated spikes across the 2 s trial. After the forward model, however, the burst would be transformed into an event with a magnitude many times greater than the events associated with isolated spikes, due to the nonlinear relationship between spike counts and the resulting calcium-dependent fluorescence. This effect can be seen in stimulus-locked raster plots for the same neuron before and after applying the forward model (*Figure 5C*).

What effects does this transformation have on neurons' inferred functional properties? Applying the forward model plus event extraction to the ephys data did not systematically alter the fraction of

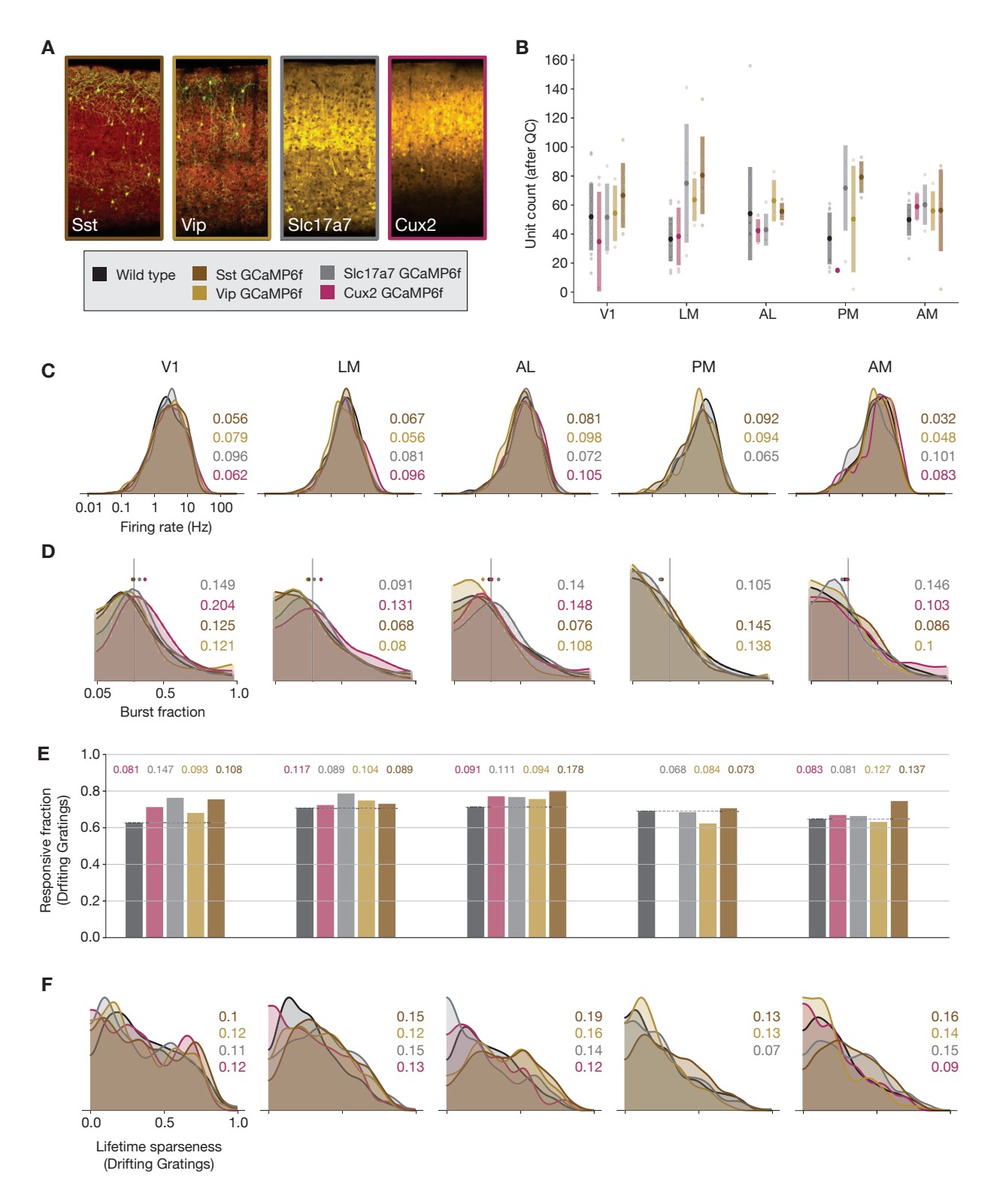

**Figure 4.** Comparing responses across GCaMP-expressing mouse lines. (A) GCaMP expression patterns for the four lines used for ephys experiments, based on two-photon serial tomography (TissueCyte) sections from mice used in the imaging pipeline. (B) Unit yield (following QC filtering) for five areas and five genotypes. Error bars represent standard deviation across experiments; each dot represents a data point from one experiment. (C) Distribution of firing rates for neurons from each mouse line, aggregated across experiments. (D) Distribution of burst fraction (fraction of all spikes that

*Figure 4 continued on next page*

*Figure 4 continued*

participate in bursts) for neurons from each mouse line, aggregated across experiments. Dots represent the median of each distribution, shown in relation to a reference value of 0.3. (E) Fraction of neurons deemed responsive to drifting gratings, grouped by genotype. (F) Distribution of lifetime sparseness in response to a drifting grating stimulus, grouped by genotype. In panels (C–F), colored numbers indicate the Jensen–Shannon distance between the wild-type distribution and the distributions of the four GCaMP-expressing mouse lines. Area PM does not include data from Cux2 mice, as it was only successfully targeted in one session for this mouse line.

responsive units in the dataset. While 8% of neurons switched from being responsive to drifting gratings to unresponsive, or vice versa, they did so in approximately equal numbers (*Figure 5D*). The forward model did not improve the match between the distributions of response reliabilities (our responsiveness metric) for any stimulus type (*Figure 5E*). The forward model similarly had a negligible impact on preference metrics; for example, only 14% of neurons changed their preferred temporal frequency after applying the forward model (*Figure 5F*), and the overall distribution of preferred temporal frequencies still matched that from the imaging experiments (*Figure 5G*). In contrast, nearly all neurons increased their selectivity after applying the forward model (*Figure 5H*). Overall, the distribution of lifetime sparseness to drifting gratings became more similar to—but still did not completely match—the imaging distribution across all areas (*Figure 5I*). The average Jensen–Shannon distance between the ephys and imaging distributions was 0.41 before applying the forward model, compared to 0.14 afterward (mean bootstrapped distance between the sub-samples of the imaging distribution = 0.064; $p < 0.001$ for all areas, since 1000 bootstrap samples never exceeded the true Jensen–Shannon distance; see Materials and methods for details). These results imply that the primary effects of the forward model—providing a supralinear boost to the 'amplitude' of spike bursts, and thresholding out single spike events—can account for baseline differences in selectivity, but not responsiveness, between ephys and imaging.

To assess whether the discrepancies between the imaging and ephys distributions of responsiveness and selectivity metrics could be further reduced by using a different set of forward model parameters, we brute-force sampled 1000 different parameter combinations for one ephys session, using 10 values each for amplitude, decay time, and noise level (*Figure 5—figure supplement 1A*), spanning the entire range of parameters calibrated on the experimental imaging data. The fraction of responsive neurons did not change as a function of forward model parameters, except for the lowest values of amplitude and noise level, where it decreased substantially (*Figure 5—figure supplement 1B*). This parameter combination (A $\leq$ 0.0015, sigma $\leq$ 0.03) was observed in less than 1% of actual neurons recorded with two-photon imaging, so it cannot account for differences in responsiveness between the two modalities. Both the difference between the median lifetime sparseness for imaging and ephys, as well as the Jensen–Shannon distance between the full ephys and imaging lifetime sparseness distributions, were near the global minimum for the parameter values we initially used (*Figure 5—figure supplement 1C,D*).

It is conceivable that the inability of the forward model to fully reconcile differences in responsiveness and selectivity was due to the fact that we applied the same parameters across all neurons of the ephys dataset, without considering their genetically defined cell type. To test for cell-type-specific differences in forward model parameters, we examined the distributions of amplitude, decay time, and noise level for individual excitatory Cre lines used in the imaging dataset. The distributions of parameter values across genotypes were largely overlapping, with the exception of increasing noise levels for some of the deeper populations (e.g. Tlx3-Cre in layer 5, and Ntsr1-Cre_GN220 in layer 6) and an abundance of low-amplitude neurons in the Fezf2-CreER population (*Figure 5—figure supplement 2A*). Given that higher noise levels and lower amplitudes did not improve the correspondence between the ephys and imaging metric distributions, we concluded that selecting parameter values for individual neurons based on their most likely cell type would not change our results. Furthermore, we saw no correlation between responsiveness or selectivity metrics in imaged neurons and their calibrated amplitude, decay time, or noise level (*Figure 5—figure supplement 2B–E*).

## Effect of ephys selection bias

We next sought to determine whether electrophysiology's well-known selection bias in favor of more active neurons could account for the differences between modalities. Whereas calcium

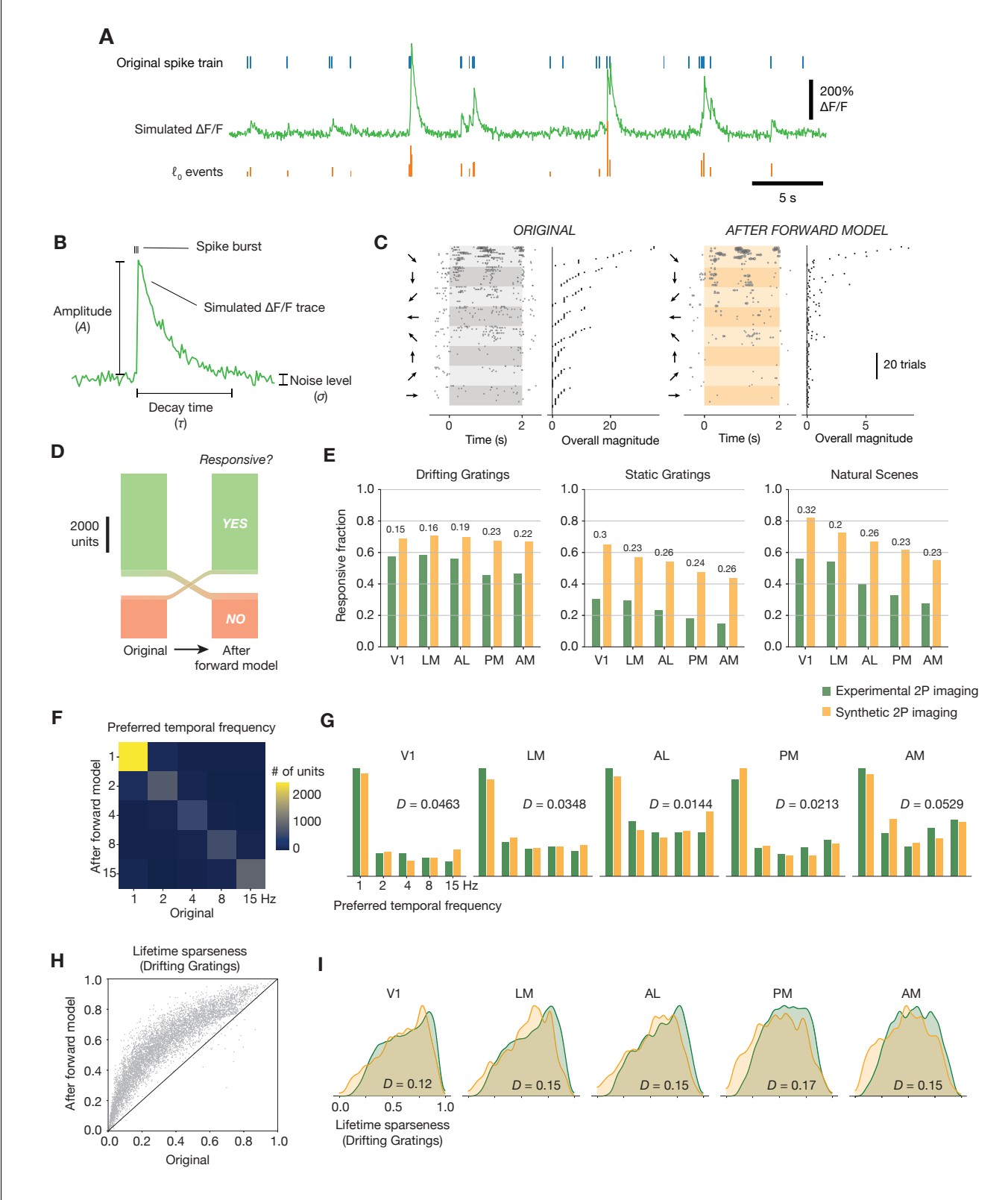

**Figure 5.** Effects of the spikes-to-calcium forward model. (**A**) Example of the forward model transformation. A series of spike times (top) is converted to a simulated ΔF/F trace (middle), from which ℓ₀-regularized events are extracted using the same method as for the experimental imaging data (bottom). (**B**) Overview of the three forward model parameters that were fit to data from the Allen Brain Observatory two-photon imaging experiments. (**C**) Raster plot for one example neuron, before and after applying the forward model. Trials are grouped by drifting grating direction and ordered by response

*Figure 5 continued on next page*

*Figure 5 continued*

magnitude from the original ephys data. The magnitudes of each trial (based on ephys spike counts or summed imaging event amplitudes) are shown to the right of each plot. (**D**) Count of neurons responding (green) or not responding (red) to drifting gratings before or after applying the forward model. (**E**) Fraction of neurons deemed responsive to each of three stimulus types, for both synthetic fluorescence traces (yellow) and true imaging data (green). Numbers above each pair of bars represent the Jensen–Shannon distance between the full distribution of responsive trial fractions for each stimulus/area combination. (**F**) 2D histogram of neurons' preferred temporal frequency before and after applying the forward model. (**G**) Distribution of preferred temporal frequencies for all neurons in five different areas. The Jensen–Shannon distance between the synthetic imaging and true imaging distributions is shown for each plot. (**H**) Comparison between measured lifetime sparseness in response to drifting grating stimulus before and after applying the forward model. (**I**) Distributions of lifetime sparseness in response to a drifting grating stimulus for all neurons in five different areas. The Jensen–Shannon distance between the synthetic imaging and true imaging distributions is shown for each plot.

The online version of this article includes the following figure supplement(s) for figure 5:

**Figure supplement 1.** Sampling the entire space of forward model parameters.

**Figure supplement 2.** Relationship between forward model parameters, genotype, and functional metrics.

imaging can detect the presence of all neurons in the field of view that express a fluorescent indicator, ephys cannot detect neurons unless they fire action potentials. This bias is exacerbated by the spike sorting process, which requires a sufficient number of spikes in order to generate an accurate template of each neuron's waveform. Spike sorting algorithms can also mistakenly merge spikes from nearby neurons into a single 'unit' or allow background activity to contaminate a spike train, especially when spike waveforms generated by one neuron vary over time, for example due to the adaptation that occurs during a burst. These issues all result in an apparent activity level increase in ephys recordings. In addition, assuming a 50-μm 'listening radius' for the probes (radius of half-cylinder around the probe where the neurons' spike amplitude is sufficiently above noise to trigger detection) (**Buzsáki, 2004**; **Harris et al., 2016**; **Shoham et al., 2006**), the average yield of 116 regular-spiking units/probe (prior to QC filtering) would imply a density of 42,000 neurons/mm$^3$, much lower than the known density of ~90,000 neurons/mm$^3$ for excitatory cells in mouse visual cortex (**Erö et al., 2018**).

If the ephys dataset is biased toward recording neurons with higher firing rates, it may be more appropriate to compare it with only the most active neurons in the imaging dataset. To test this, we systematically *increased* the event rate threshold for the imaged neurons, so the remaining neurons used for comparison were always in the upper quantile of mean event rate. Applying this filter increased the overall fraction of responsive neurons in the imaging dataset, such that the experimental imaging and synthetic imaging distributions had the highest similarity when between 7 and 39% of the most active imaged neurons were included (V1: 39%, LM: 34%, AL: 25%, PM: 7%, AM: 14%) (**Figure 6A**). This indicates that more active neurons tend to be more responsive to our visual stimuli, which could conceivably account for the discrepancy in overall responsiveness between the two modalities. However, applying this event rate threshold actually increased the differences between the selectivity distributions, as the most active imaged neurons were also more selective (**Figure 6B**). Thus, sub-selection of imaged neurons based on event rate was not sufficient to fully reconcile the differences between ephys and imaging. Performing the same analysis using sub-selection based on the coefficient of variation, an alternative measure of response reliability, yielded qualitatively similar results (**Figure 6—figure supplement 1**).

If the ephys dataset includes spike trains that are contaminated with spurious spikes from one or more nearby neurons then it may help to compare our imaging results only to the least contaminated neurons from the ephys dataset. Our initial QC process excluded units with an inter-spike interval (ISI) violations score (**Hill et al., 2011**, see Materials and methods for definition) above 0.5, to remove highly contaminated units, but while the presence of refractory period violations implies contamination, the absence of such violations does not imply error-free clustering, so some contamination may remain. We systematically *decreased* our tolerance for ISI-violating ephys neurons, so the remaining neurons used for comparison were always in the lower quantile of contamination level. For the most restrictive thresholds, where there was zero detectable contamination in the original spike trains (ISI violations score = 0), the match between the synthetic imaging and experimental imaging selectivity and responsiveness distributions was maximized (**Figure 6C,D**). This indicates that, unsurprisingly, contamination by neighboring neurons (as measured by ISI violations score) reduces selectivity and increases responsiveness. Therefore, the inferred functional properties are

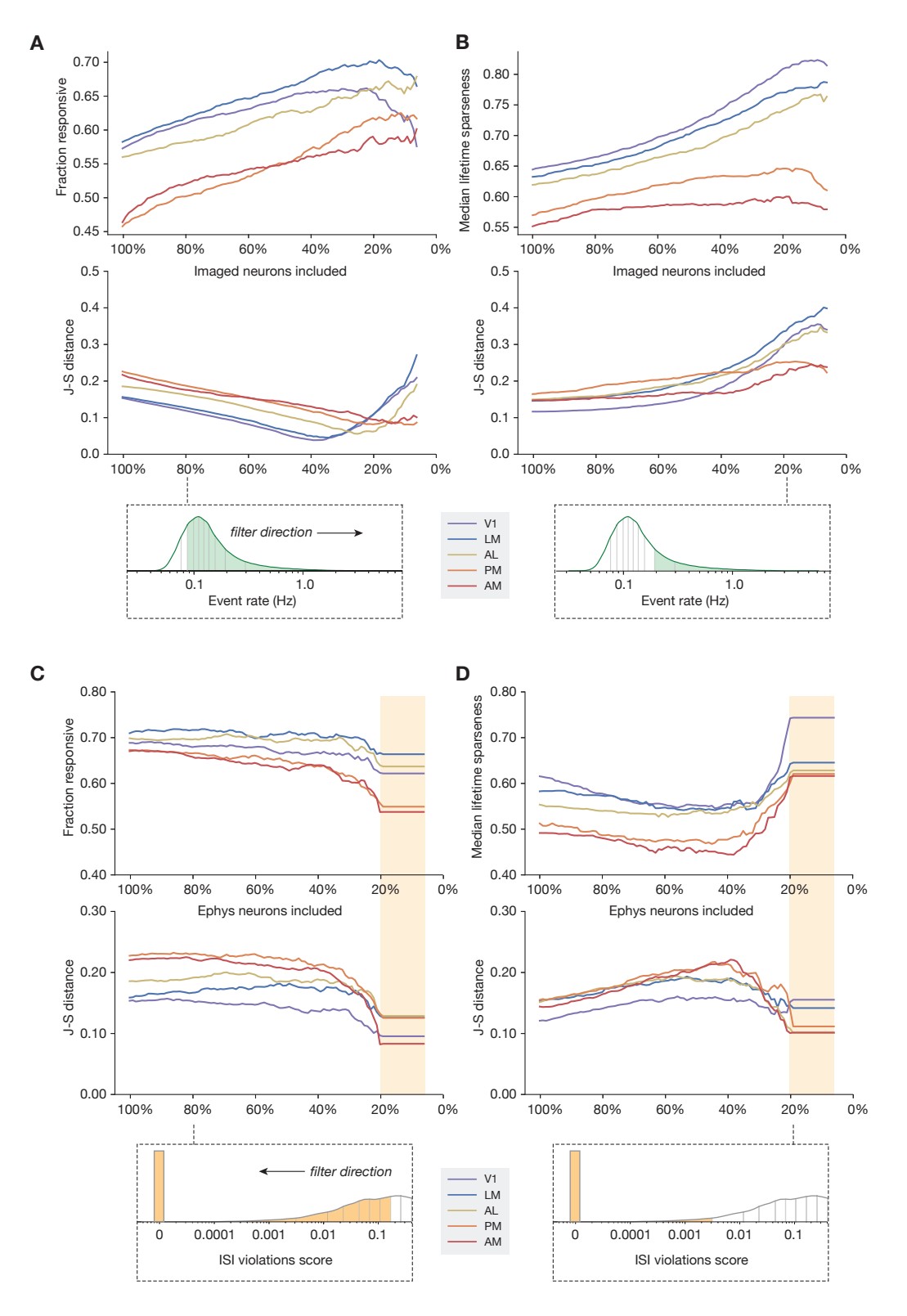

**Figure 6.** Sub-sampling based on event rate or ISI violations. (**A**) *Top:* Change in fraction of neurons responding to drifting gratings for each area in the imaging dataset, as function of the percent of imaged neurons included in the comparison. *Middle:* Jensen–Shannon distance between the synthetic imaging and true imaging response reliability distributions. *Bottom:* Overall event rate distribution in the imaging dataset. As the percentage of included neurons decreases (from left to right), more and more neurons with low activity rates are excluded. (**B**) Same as A, but for drifting gratings

*Figure 6 continued on next page*

*Figure 6 continued*

lifetime sparseness (selectivity) metrics. (C) Top: Change in fraction of neurons responding to the drifting gratings stimulus for each area in the ephys dataset, as function of the percent of ephys neurons included in the comparison. Middle: Jensen–Shannon distance between the synthetic imaging and true imaging response reliability distributions. Bottom: Overall ISI violations score distribution in the ephys dataset. As the percent of included neurons decreases (from right to left), more and more neurons with high contamination levels are excluded. Yellow shaded area indicates the region where the minimum measurable contamination level (ISI violations score = 0) is reached. (D) Same as C, but for drifting gratings lifetime sparseness (selectivity) metrics.

The online version of this article includes the following figure supplement(s) for figure 6:

**Figure supplement 1.** An alternative metric of response robustness.

most congruent across modalities when the ephys analysis includes a stringent threshold on the maximum allowable contamination level.

## Results across stimulus types

The previous results have primarily focused on the drifting gratings stimulus, but we observe similar effects for all of the stimulus types shared between the imaging and ephys datasets. *Figure 7* summarizes the impact of each transformation we performed, either before or after applying the forward model, for drifting gratings, static gratings, natural scenes, and natural movies.

Across all stimulus types, the forward model had very little impact on responsiveness. Instead, sub-selecting the most active neurons from our imaging experiments using an event-rate filter rendered the shape of the distributions the most similar. For the stimulus types for which we could measure preference across small number of categories (temporal frequency of drifting gratings and spatial frequency of static gratings), no data transformations were able to improve the overall match between the ephys and imaging distributions, as they were already very similar in the baseline comparison. For selectivity metrics (lifetime sparseness), applying the forward model played the biggest role in improving cross-modal similarity, although there was a greater discrepancy between the resulting distributions for static gratings, natural scenes, and natural movies than there was for drifting gratings. Filtering ephys neurons based on ISI violations further reduced the Jensen–Shannon distance, but it still remained well above zero. This indicates that the transformations we employed could not fully reconcile observed differences in selectivity distributions between ephys and imaging.

## Lessons for future comparisons

Our study shows that population-level functional metrics computed from imaging and electrophysiology experiments can display systematic biases. What are the most important takeaways that should be considered for those performing similar comparisons?

## Preference metrics are similar across modalities

At least for the cell population we considered (putative excitatory neurons from all layers of visual cortex), preference metrics (such as preferred temporal frequency, preferred spatial frequency, and preferred direction) were largely similar between imaging and electrophysiology (*Figure 2F*, *Figure 2—figure supplement 1A–C*). Because these are categorical metrics defined as the individual condition (out of a finite set) that evokes the strongest mean response, they are robust to the choice of calcium event extraction method and also remain largely invariant to the application of a spikes-to-calcium forward model to electrophysiology data. One caveat to keep in mind is that when imaging from more specific populations (e.g. using a transgenic line that limits imaging to a specific subtype of neuron in a specific layer), electrophysiology experiments may yield conflicting preference metrics unless the sample is carefully matched across modalities (e.g. by genetically tagging electrically recorded neurons with a light-sensitive opsin).

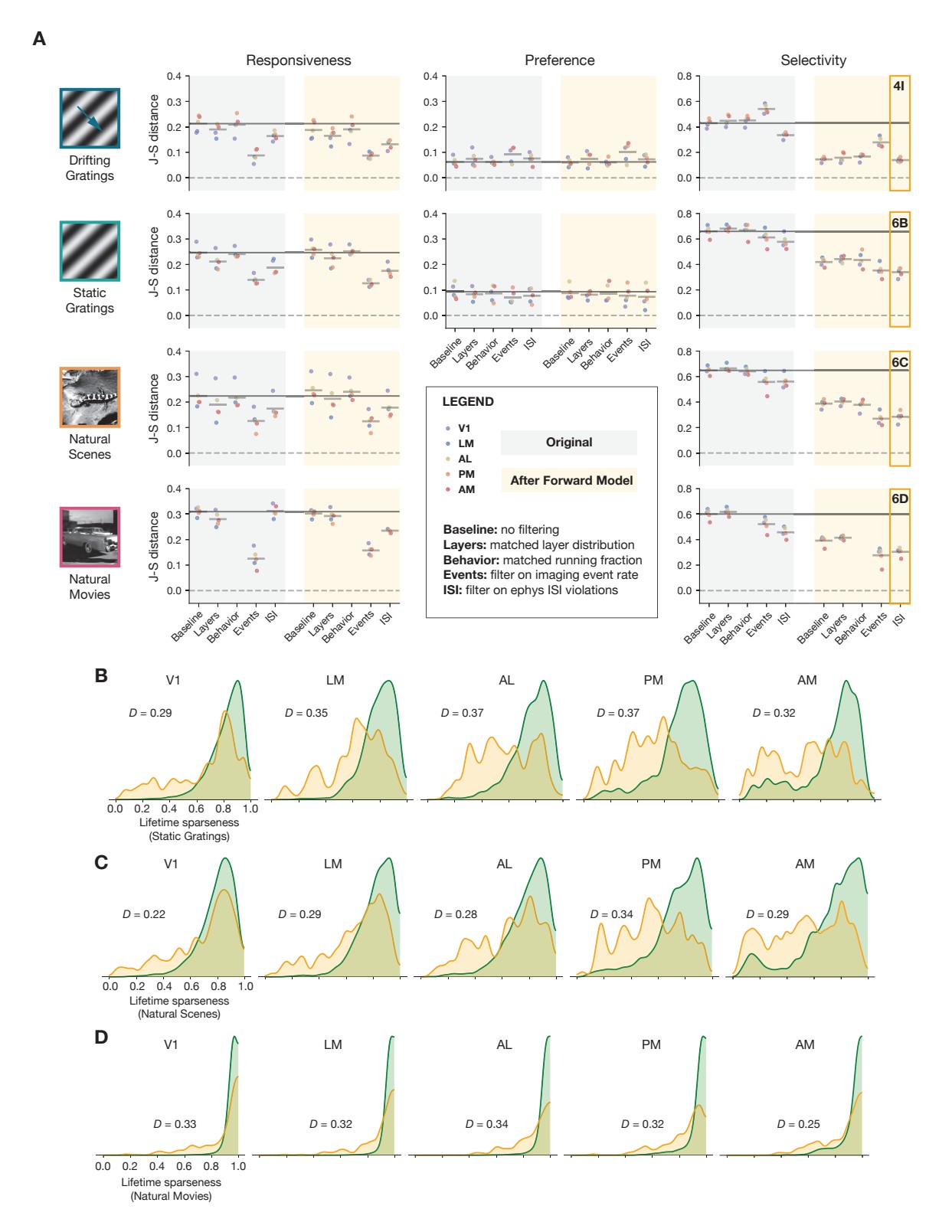

**Figure 7.** Summary of results for all stimulus types. (**A**) Jensen–Shannon distance between the ephys and imaging distributions for three metrics and four stimulus classes. Individual dots represent visual areas, and horizontal lines represent the mean across areas. Each column represents a different post-processing step, with comparison performed on either the original ephys data (gray), or the synthetic fluorescence data obtained by passing experimental ephys data through a spikes-to-calcium forward model (yellow). (**B**) Full distributions of lifetime sparseness in response to static gratings

*Figure 7 continued on next page*

*Figure 7 continued*

for experimental imaging (green) and synthetic imaging (yellow). (**C**) same as B, but for natural scenes. (**D**), same as B and C, but for natural movies. The distribution for drifting gratings is shown in ***Figure 5I***.

## Differences in responsiveness metrics largely stem from ephys selection bias

In our original comparison, a larger fraction of ephys units were found to be responsive to every stimulus type (***Figure 2E***); this did not change after applying a spikes-to-calcium forward model (***Figure 5D,E***). We believe this is primarily due to two factors:

1. Extracellular electrophysiology cannot detect neurons if they fire few or no action potentials within the sampling period. Unresponsive neurons with low backround firing rates fall into this category, and are therefore not counted in the 'yield' of an ephys experiment.
2. Most or all ephys 'units' include some fraction of contaminating spikes from nearby neurons with similar waveform shapes. Contamination is more likely to occur during periods of high waveform variability, for example during burst-dependent spike amplitude adaptation. Because bursts are prevalent when a neuron responds to a stimulus, stimulus-evoked spikes are the ones that are most likely to contaminate the spike train of an unresponsive cell.

How should these biases be accounted for? Most importantly, when comparing responsiveness, or analyses that build on responsiveness, between ephys and imaging experiments, only the cleanest, least contaminated ephys units should be included (based on their ISI violations or another purity metric). In addition, one can reduce differences in responsiveness by filtering out the least active neurons from imaging experiments, or simply by using a higher responsiveness threshold for ephys than for imaging. For example, one could use a sliding threshold to find the point where the overall rate of responsive neurons is matched between the two modalities, and perform subsequent comparisons using this threshold. It should also be noted that the method of calcium event detection can also affect responsiveness metrics; with a more permissive event detection threshold, for instance, the population appears more responsive (***Figure 3G***). However, it is clear that a lower threshold leads to a higher fraction of false-positive events, as is shown using ground truth data (***Figure 3D***), and this increases the probability that noise in the underlying fluorescence will contaminate the results. As background fluorescnece is most variable when nearby neurons or processes respond to their respective preferred stimulus condition, additionally detected events are likely to be an overfitting of stimulus-correlated noise .

## Selectivity metrics are highly sensitive to the parameters used for calcium event extraction

Differences in selectivity (or tuning curve sharpness) are the most difficult to compare across modalities. This is because most commonly used selectivity metrics take into account the ratio between the peak and baseline response, and the relative size of these responses is highly influenced by the rate and size of 'background' events. When counting spikes in electrophysiology, the largest and smallest responses typically fall within the same order of magnitude; with imaging, however, calcium event amplitudes can easily vary over several orders of magnitude. In addition, the specific method used for calcium event detection can have a big impact on background event rate. Because these events always have a positive amplitude, events detected around the noise floor cannot be cancelled out by equivalently sized negative events. In principle, one could try to match selectivity across modalities by tuning the parameters of the event extraction algorithm. However, this is not recommended, because it obfuscates real biases in the data (such as the sparsifying effect of calcium indicators) and can lead to inconsistencies (e.g. it fails to consistently match both selectivity and responsiveness across stimulus classes). Instead, as a more principled way to compare selectivity between imaging and ephys experiments, we recommend the use of a spikes-to-calcium forward model, with the same event extraction algorithm applied to both the real and the synthetic calcium traces (***Figure 5I***, ***Figure 7B–D***).

## Inter-modal differences in running speed, laminar sampling patterns, and transgenic mouse lines do not substantially bias functional metrics

Our ephys recordings included a higher fraction of neurons from layer 5 than our imaging experiments (*Figure 1F*), while mice from imaging experiments were less active runners (*Figure 2—figure supplement 3A*). Furthermore, typical ephys experiments do not use transgenic mice that express calcium indicators, while this is common for imaging. Correcting these biases did not appreciably change the population-level functional metrics. However, it must be noted that we used nearly identical behavioral apparatuses, habituation protocols, and stimulus sets between modalities. When comparing across studies with methods that were not as carefully matched, behavioral differences may have a larger influence on the results.

## Interpreting higher order analyses in light of our findings

The differences in responsiveness and selectivity metrics computed from the ephys and imaging datasets suggest that functional properties of neurons in the mouse visual cortex can appear to be dependent on the choice of recording modality. These effects extend beyond responsiveness and selectivity, as higher order analyses often build on these more fundamental metrics. Here, we focus on a representative example, a functional classification scheme based on responses to four stimulus types (drifting gratings, static gratings, natural scenes, and natural movies), which, in our previous work, revealed distinct response classes in the imaging dataset (*de Vries et al., 2020*). We aim to illustrate, using this recent example, that the effect of recording modality on responsiveness goes far beyond merely impacting how many neurons might be included in an analysis but may also propagate to conclusions we draw about the functional properties of neuronal (sub)populations. Our original study, based on imaging data alone, revealed that only ~10% of the neurons in the imaging dataset responded reliably to all four stimuli, while the largest class of neurons contained those that did not respond reliably to any of the stimuli. This classification result suggested that many neurons in the mouse visual cortex are not well described by classical models and may respond to more intricate visual or non-visual features. Here, we perform an analogous analysis on the ephys dataset to show how this classification is impacted by modality specific biases.

As in our original study, we performed unsupervised clustering on the 4 x *N* matrix of response reliabilities for each unit's preferred condition for each stimulus class, where *N* represents the number of units (*Figure 8A*). The resulting clusters were assigned class labels based on whether their mean response reliability was above an initial threshold of 25% (to match the percentage we used in our previous analysis of the imaging dataset). After labeling each cluster, we calculated the fraction of units belonging to each functional class. We averaged over 100 different clustering initializations to obtain the average class membership of units across runs (*Figure 8B*).

Running this analysis on the ephys dataset 'naively', seemed to reveal a very different landscape of functional classes. In stark contrast to the published imaging results, for ephys, the largest class (~40% of the units) contained units that responded reliably to all four classes of stimuli, while the class that did not respond reliably to any of the stimuli was empty. This is consistent with the observation that responsiveness is higher in the ephys dataset for each stimulus type (*Figure 2F*). To account for this bias, we systematically raised the responsiveness threshold used to group clusters into functional classes for the ephys dataset and found that the distribution of classes became more similar to the distribution of classes for the imaging dataset (*Figure 8C*). A threshold of 40% response reliability for the ephys dataset minimized the Jensen–Shannon distance between the distributions, rendering the class assignments in ephys remarkably similar to those for the imaging dataset (*Figure 8E*). The class labels for each neuron reflect the pattern of cross-stimulus response reliability, and as such provide an indication of its 'meta-preference' for different stimulus types. Thus, once we account for the generally higher levels of responsiveness seen in the ephys dataset, we observe similar meta-preferences and thus functional organization as for the imaging dataset. This example highlights how recording-modality-specific biases can affect higher-order conclusions about functional properties, and how a fundamental understanding of such biases can be leveraged to explain and resolve apparent contradictions.

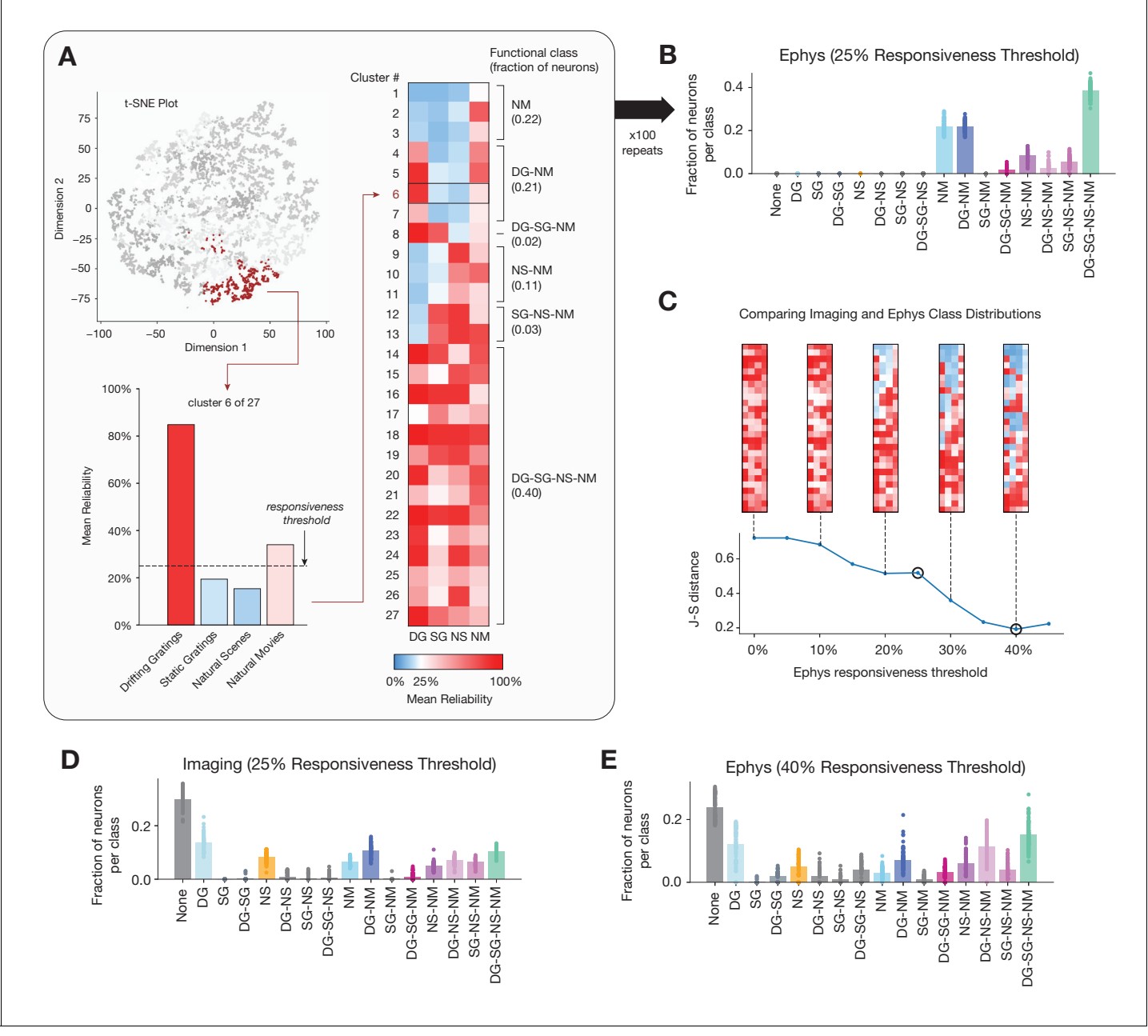

**Figure 8.** Identifying functional classes based on response reliability. (A) Example clustering run for all neurons in the ephys dataset. First, unsupervised clustering is performed on the matrix of response reliabilities to four stimulus types. Colors on the t-SNE plot (used for visualization purposes only) represent different cluster labels, with one cluster highlighted in brown. Next, the neurons in each cluster are used to compute the mean reliability for each stimulus type. The mean reliabilities for all 27 clusters are shown in the heatmap, with the 25% responsiveness threshold indicated by the transition from blue to red. Each cluster is assigned to a class based on the stimulus types for which its mean reliability is above threshold. (B) Strip plots representing the fraction of neurons belonging to each class, averaged over 100 repeats of the Gaussian mixture model for ephys (N = 11,030 neurons), at a 25% threshold level. (C) Jensen–Shannon distance between the ephys and imaging class membership distributions, as a function of the ephys responsiveness threshold. The points plotted in panels B and E are circled. Heat maps show the mean reliabilities relative to threshold for all 27 clusters shown in panel A. (D) Strip plots representing the fraction of neurons belonging to each class, averaged over 100 repeats of the Gaussian mixture model for imaging (N = 20,611 neurons), at a 25% threshold level. (E) Same as B, but for the 40% threshold level that minimizes the difference between the imaging and ephys distributions.

# Discussion

In this study, we have compared response metrics derived from mouse visual cortex excitatory populations collected under highly standardized conditions, but using two different recording modalities. Overall, we observe similar stimulus preferences across the two datasets (e.g. preferred temporal frequencies within each visual area), but we see systematic differences in responsiveness and selectivity. Prior to any attempt at reconciliation, electrophysiological recordings showed a higher fraction of units with stimulus-driven activity, while calcium imaging showed higher selectivity (sharper tuning) among responsive neurons.

Our comparison of 2P event detection methods showed that the rate of small-amplitude events can influence inferred functional metrics. While the prevalence of false positives in ground truth data has been analyzed previously (*Theis et al., 2016*; *Rupprecht et al., 2021*), this metric is not typically used to optimize event detection parameters. Instead, correlation with ground truth rate is preferred (*Berens et al., 2018*; *Pachitariu et al., 2018*). However, this measure does not account for changes in background rate associated with the prevalence of small-amplitude events. These events always have a positive magnitude, are often triggered by noise, and can dramatically affect measured selectivity levels. In fact, most neurons have selectivity levels near zero when all events detected by non-negative deconvolution are included in the analysis (*Figure 3F*). Measured responsiveness, on the other hand, was highest when all events were included (*Figure 3G*). While this could indicate a greater sensitivity to true underlying spikes, it could also result from contamination by fluctuations in background fluorescence during the visual stimulation interval. Because of confounds such as these, we found it more informative to carry out our comparison on ephys data that has been transformed by the forward model and processed using the same event extraction method as the experimental imaging data.

Notably, the forward model boosted selectivity of the ephys data due to the sparsifying effect of calcium dynamics and the exact $\ell_0$ event extraction step. As large, burst-dependent calcium transients can have amplitudes several orders of magnitude above the median amplitude (*Figure 2B*), this causes the response to the preferred stimulus condition to be weighted more heavily in selectivity calculations than non-preferred conditions. Similarly, isolated spikes during non-preferred conditions can be virtually indistinguishable from noise when viewed through the lens of the forward model. When the same trials are viewed through the lens of electrophysiology, however, spike counts increase more or less linearly, leading to the appearance of lower selectivity.

Unexpectedly, the forward model did not change responsiveness metrics. We initially hypothesized that the lower responsiveness in the imaging dataset was due to the fact that single-spike events are often not translated into detectable calcium transients; in ground truth recordings, around 75% of 100 ms bins with actual spikes do not result in any events detected by the exact $\ell_0$ method with the default sparsity constraint (*Huang et al., 2021*). Instead, our observation of unchanged responsiveness following the forward model suggests that differences between modalities are more likely due to electrophysiological sampling bias—that is extracellular recordings missing small or low-firing rate neurons, or merging spike trains from nearby cells. In order to reconcile some of these differences, we could either apply a very strict threshold on ISI violations score to the ephys dataset, or remove between 61 and 93% of the least active neurons from the imaging dataset (*Figure 6*). This should serve as a cautionary tale for anyone estimating the fraction of neurons in an area that appear to increase their firing rate in response to environmental or behavioral events. Without careful controls, contamination from other neurons in the vicinity can make this fraction appear artificially high. While it may be possible to minimize these impurities with improved spike-sorting algorithms, there will always be neurons that even the best algorithms will not be able to distinguish in the face of background noise.

Differences in laminar sampling and running behavior between the two modalities had almost no effect in our comparisons. The transgenic expression of GCaMP6f in specific neural populations also did not impact the distributions of functional metrics. Finally, the initial parameters chosen for the forward model produced metric distributions that were close to the optimum match over the realistic parameter space. Therefore, we conclude that the primary contribution to differences in the considered ephys and imaging metrics comes from (1) the intrinsic nature of the spikes-to-calcium transfer function (2) the selection bias of extracellular electrophysiology recordings.

Even after accounting for these known factors to the best of our ability, the overall population from our imaging experiments still displayed higher selectivity than its ephys counterpart (*Figure 5I*, *Figure 8B–D*). What could account for these remaining differences? One possibility is that there may be residual undetected contamination in our ephys recordings. An ISI violations score of 0 does not guarantee that there is no contamination, just that we are not able to measure it using this metric. Sampling the tissue more densely (i.e. increasing the spatial resolution of spike waveforms) or improving spike sorting methods could reduce this issue. Another possibility is that 'missed' spikes—especially those at the end of a burst—could result in reduced amplitudes for the simulated calcium transients. In addition, if the in vivo spikes-to-calcium transfer function is non-stationary, there could be stimulus-dependent changes in calcium concentration that are not captured by a forward model that takes a spike train as its only input. Simultaneous cell-attached electrophysiology and two-photon imaging experiments have demonstrated the existence of 'prolonged depolarization events' (PDEs) in some neurons that result in very large increases in calcium concentration, but that are indistinguishable from similar burst events in extracellular recordings (*Ledochowitsch et al., 2019*).

One potential limitation of our approach is that we have only imaged the activity of transgenically expressed calcium indicators, rather than indicators expressed using a viral approach. In addition, the vast majority of our imaging data comes from mice expressing GCaMP6f, with only a small number of GCaMP6s neurons recorded. While we would ideally want to perform the same experiments with viral expression and GCaMP6s, this would require an expensive multi-year effort of similar magnitude to the one that produced our existing imaging dataset. Instead, we have chosen to simulate the effects of these alternative conditions. In our analysis of the forward model parameter sweep, functional metrics remain relatively constant for a wide range of amplitude and decay time parameters (*Figure 5—figure supplement 1*). The full range of this sweep includes decay times that are consistent with GCaMP6s, and event amplitudes that are consistent with viral expression.

The forward models currently available in the literature (*Deneux et al., 2016*; *Wei et al., 2019*) are of comparable power, in that their most complex instantiations allow for non-instantaneous fluorescence rise as well as for a non-linear relationship between calcium concentration and fluorescence. To the best of our knowledge, none of these forward models explicitly model nonstationarities in the spike-to-calcium transfer function. Moreover, all currently available models suffer from the drawback that fits to simultaneously recorded ground truth data yield significant variance in model parameters across neurons (*Ledochowitsch et al., 2019*; *Wei et al., 2019*). We strove to mitigate this shortcoming by showing that brute-force exploration of MLSpike model parameter space could not significantly improve the match between real and synthetic imaging data.

Another potential confound is elevated activity around an implanted probe, which was characterized in a recent study (*Eles et al., 2018*). By performing calcium imaging around a silicon probe that was recently introduced into the brain, the authors found increased intracellular calcium lasting for at least 30 min after implantation. If this is true in our Neuropixels recordings, it could at least partially account for the higher overall firing rates and responsiveness in electrophysiology compared to imaging. Careful simultaneous measurements will be required in order to account for this relative activity increase.

While results obtained from ephys and imaging are sometimes treated as if they were interchangeable from a scientific standpoint, in actuality they each provide related but fundamentally different perspectives on the underlying neural activity. Extracellular electrophysiology tells us—with sub-millisecond temporal resolution—about a neuron's spiking output. Calcium imaging doesn't measure the outgoing action potentials directly, but rather the impact of input and output signals on a neuron's internal state, in terms of increases in calcium concentration that drive various downstream pathways (*West et al., 2001*). While voltage-dependent fluorescent indicators may offer the best of both worlds, there are substantial technical hurdles to employing them on comparably large scales (*Peterka et al., 2011*). Thus, in order to correctly interpret existing and forthcoming datasets, we must account for the inherent biases of these two recording modalities.

A recent study comparing matched populations recorded with electrophysiology or imaging emphasized differences in the temporal profiles of spike trains and calcium-dependent fluorescence responses (*Wei et al., 2019*). The authors found that event-extraction algorithms that convert continuous ΔF/F traces to putative spike times could not recapitulate the temporal profiles measured with electrophysiology; on the other hand, a forward model that transformed spike times to synthetic ΔF/

F traces could make their electrophysiology results appear more like those from the imaging experiments. Their conclusions were primarily based on metrics derived from the evolution of firing rates or ΔF/F ratios over the course of a behavioral trial. However, there are other types of functional metrics that are not explicitly dependent on temporal factors, such as responsiveness, which cannot be reconciled using a forward model alone.

It is worth emphasizing that we are not suggesting that researchers use the methods we describe here to attempt to make all of their imaging data more similar to electrophysiological data, or vice versa. Since no one single method is intrinsically superior to the other, doing so would merely introduce additional biases. Instead, we recommend that readers examine how sensitive their chosen functional metrics, and, by extension, their derived scientific conclusions are to (1) signal contamination during spike detection and sorting (for electrophysiology data), (2) application of a spike-to-calcium forward model (for electrophysiology data), (3) filtering by event rate (for imaging data), and (4) false positives introduced during event detection (for imaging data). If the chosen functional metrics are found to be largely insensitive to the above transformations, then results can be compared directly across studies that employ different recording modalities. Otherwise, the sensitivity analysis can be used as a means of establishing bounds on the magnitude and direction of expected modality-related discrepancies.

More work is needed to understand the detailed physiological underpinning of the modality-specific differences we have observed. One approach, currently underway at the Allen Institute and elsewhere, is to carry out recordings with extremely high-density silicon probes (Neuropixels Ultra), over 10x denser (in terms of the number of electrodes per unit of area) than the Neuropixels 1.0 probes used in this study. Such probes can capture each spike waveform with 100 or more electrodes, making it easier to disambiguate waveforms from nearby neurons, and making it less likely that neurons with small somata would evade detection or that their waveforms would be mistaken for those of other units. These experiments should make it easier to quantify the selection bias of extracellular electrophysiology, as well as the degree to which missed neurons and contaminated spike trains have influenced the results of the current study. In addition, experiments in which silicon probes are combined with two-photon imaging—either through interleaved sampling, spike-triggered image acquisition, or improved artifact removal techniques—could provide more direct ground-truth information about the relationship between extracellular electrophysiology and calcium imaging.

Overall, our comparison highlights the value of large-scale, standardized datasets. The fact that functional metrics are sensitive not only to experimental procedures but also to data processing steps and cell inclusion criteria (*Mesa et al., 2021*), makes it difficult to directly compare results across studies. Having access to ephys and imaging datasets collected under largely identical conditions allowed us to rule out a number of potential confounds, such as laminar sampling bias and inter-modality behavioral differences. And due to the technical difficulties of scaling simultaneous ephys/imaging experiments, these will, for the foreseeable future, continue to complement and validate large-scale unimodal datasets, rather than replace them.

Ultimately, the goal of this work is not to establish the superiority of any one recording modality in absolute terms, since their complementary strengths ensure they will each remain essential to scientific progress for many years to come. Instead, we want to establish guidelines for properly interpreting the massive amounts of data that have been or will be collected using either modality. From this study, we have learned that extracellular electrophysiology likely overestimates the fraction of neurons that elevate their activity in response to visual stimuli, in a manner that is consistent with the effects of selection bias and contamination. The apparent differences in selectivity underscore the fact that one must carefully consider the impact of data processing steps (such as event extraction from fluorescence time series), as well as what each modality is actually measuring. Selectivity metrics based on spike counts (the neuron's outputs) will almost always be lower than selectivity metrics based on calcium concentrations (the neuron's internal state). Even with this in mind, however, we cannot fully reproduce the observed levels of calcium-dependent selectivity using spike times alone—suggesting that a neuron's internal state may contain stimulus-specific information that is not necessarily reflected in its outputs.

In summary, we have shown that reconciling results across modalities is not straightforward, due to biases that are introduced at the level of the data processing steps, the spatial characteristics of the recording hardware, and the physical signals being measured. We have attempted to account for these biases by (1) altering 2P event detection parameters, (2) applying sub-selection to account

for built-in spatial biases, and (3) simulating calcium signals via a forward model. We have shown that functional metrics are sensitive to all of these biases, which makes it difficult to determine which ones are most impactful. For example, to what degree does the bias of ephys for higher-firing-rate units stem from the recording hardware versus the spike sorting procedure? It is possible that more spatially precise sampling, or modifications to the spike sorting algorithm, will reduce this bias in the future. Similarly, how much of the 2P imaging bias stems from the calcium indicator versus the event extraction algorithm? New indicators (such as GCaMP8) may ameliorate some of these problems, as could further optimization of the event extraction step. In the end, these two modalities provide two imperfect yet complementary lenses on the underlying neural activity, and their respective strengths and limitations must be understood when interpreting the recorded activities across experiments.

## Materials and methods

### Previously released data

We used two-photon calcium imaging recordings from the Allen Brain Observatory Visual Coding dataset (*de Vries et al., 2020*; 2016 Allen Institute for Brain Science, available from observatory. brain-map.org). This dataset consists of calcium fluorescence time series from 63,521 neurons in six different cortical areas across 14 different transgenic lines. Neurons were imaged for three separate sessions (A, B, and C), each of which used a different visual stimulus set (*Figure 1E*). Our analysis was limited to neurons in five areas (V1, LM, AL, PM, and AM) and 10 lines expressing GCaMP6f in excitatory neurons, and which were present in either session A, session B, or both (total of 41,578 neurons).

We used extracellular electrophysiological recordings from the Allen Brain Observatory Neuropixels dataset (*Siegle et al., 2021*; 2019 Allen Institute for Brain Science, available from portal.brain-map.org/explore/circuits/visual-coding-neuropixels). This dataset consists of spike trains from 99,180 'units' (putative neurons with varying degrees of completeness and contamination) from 58 mice in a variety of cortical and subcortical structures. We limited our analysis to 31 sessions that used the 'Brain Observatory 1.1' stimulus set (*Figure 1E*) and units (hereafter, 'neurons') from five visual cortical areas (V1, LM, AL, PM, and AM) that displayed 'regular spiking' action potential waveforms (peak-to-trough interval > 0.4 ms). Only neurons that passed the following quality control thresholds were included: presence ratio > 0.9 (fraction of the recording session during which spikes are detected), amplitude cutoff < 0.1 (estimate of the fraction of missed spikes), and ISI violations score < 0.5 (*Hill et al., 2011*) (estimate of the relative rate of contaminating spikes). After these filtering steps, there were 5917 neurons for analysis.

### Neuropixels recordings in GCaMP6f mice

We collected a novel electrophysiology dataset from transgenic mice expressing GCaMP6f, as well as additional wild-type mice. Experiments were conducted in accordance with PHS Policy on Humane Care and Use of Laboratory Animals and approved by the Allen Institute's Institutional Animal Care and Use Committee under protocols 1409 ('A scalable data generation pipeline for creation of a mouse Cortical Activity Map'), 1706 ('Brain Observatory: Optical Physiology'), and 1805 ('Protocol for in vivo electrophysiology of mouse brain'). The procedures closely followed those described in *Siegle et al., 2021* and are summarized below.

Mice were maintained in the Allen Institute animal facility and used in accordance with protocols approved by the Allen Institute's Institutional Animal Care and Use Committee. Five genotypes were used: wild-type C57BL/6J mice purchased from Jackson Laboratories (*n* = 2) or Vip-IRES-Cre;Ai148 (*n* = 3), Sst-IRES-Cre;Ai148 (*n* = 6), Slc17a7-IRES2-Cre;Camk2a-tTA;Ai93 (*n* = 3), and Cux2-CreERT2; Camk2a-tTA;Ai93 (*n* = 3) mice bred in-house. Following surgery, mice were single-housed and maintained on a reverse 12 hr light cycle. All experiments were performed during the dark cycle.

At around age P80, mice were implanted with a titanium headframe. In the same procedure, a 5 mm diameter piece of skull was removed over visual cortex, followed by a durotomy. The skull was replaced with a circular glass coverslip coated with a layer of silicone to reduce adhesion to the brain surface.

On the day of recording (at least four weeks after the initial surgery), the glass coverslip was removed and replaced with a plastic insertion window containing holes aligned to six cortical visual

areas, identified via intrinsic signal imaging (*Garrett et al., 2014*). An agarose mixture was injected underneath the window and allowed to solidify. This mixture was optimized to be firm enough to stabilize the brain with minimal probe drift, but pliable enough to allow the probes to pass through without bending. At the end of this procedure, mice were returned to their home cages for 1–2 hr prior to the recording session.

All recordings were carried out in head-fixed mice using Neuropixels 1.0 probes (*Jun et al., 2017*; available from neuropixels.org) mounted on 3-axis stages from New Scale Technologies (Victor, NY). These probes have 383 recording sites oriented in a checkerboard pattern on a 70 µm wide x 10 mm long shank, with 20 µm vertical spacing. Data streams from each electrode were acquired at 30 kHz (spike band) and 2.5 kHz (LFP band) using the Open Ephys GUI (*Siegle et al., 2017*). Gain settings of 500x and 250x were used for the spike band and LFP band, respectively. Recordings were referenced to a large, low-impedance electrode at the tip of each probe.

Pre-processing, spike sorting, and quality control methods were identical to those used for the previously released dataset (code available at https://github.com/alleninstitute/ecephys_spike_sorting (copy archived at *Siegle, 2021*; swh:1:rev:995842e4ec67e9db1b7869d885b97317012337db) and https://github.com/MouseLand/Kilosort (copy archived at *Pachitariu, 2021*; swh:1:rev:db3a3353-d9a374ea2f71674bbe443be21986c82c)). Filtering by brain region (V1, LM, AL, PM, and AM), waveform width (>0.4 ms), and QC metrics (presence ratio > 0.9, amplitude cutoff < 0.1, ISI violations score < 0.5) yielded 5113 neurons for analysis. For all analyses except for those in *Figure 3*, neurons from this novel dataset were grouped with those from the previously released dataset, for a total of 11,030 neurons.

Neurons were registered to 3D brain volumes obtained with an open-source optical projection tomography system (https://github.com/alleninstitute/AIBSOPT, copy archived at *Nicovich, 2021*; swh:1:rev:e38af7e25651fe7517dcf7ca3d38676e3c9c211e). Brains were first cleared using a variant of the iDISCO method (*Renier et al., 2014*), then imaged with white light (for internal structure) or green light (to visualize probe tracks labeled with fluorescent dye). Reconstructed volumes were mapped to the Mouse Common Coordinate Framework (CCFv3) (*Wang et al., 2020*) by matching key points in the original brain to corresponding points in a template volume. Finally, probe tracks were manually traced and warped into the CCFv3 space, and electrodes were aligned to structural boundaries based on physiological landmarks (*Siegle et al., 2021*).

## Visual stimuli

Analysis was limited to epochs of drifting gratings, static gratings, natural scenes, or natural movie stimuli, which were shown with identical parameters across the two-photon imaging and electrophysiology experiments. Visual stimuli were generated using custom scripts based on PsychoPy (*Peirce, 2007*) and were displayed using an ASUS PA248Q LCD monitor, 1920 x 1200 pixels in size (21.93' wide, 60 Hz refresh rate). Stimuli were presented monocularly, and the monitor was positioned 15 cm from the mouse's right eye and spanned 120° x 95° of visual space prior to stimulus warping. Each monitor was gamma corrected and had a mean luminance of 50 cd/m². To account for the close viewing angle of the mouse, a spherical warping was applied to all stimuli to ensure that the apparent size, speed, and spatial frequency were constant across the monitor as seen from the mouse's perspective.

The drifting gratings stimulus consisted of a full-field sinusoidal grating at 80% contrast presented for 2 s, followed by a 1 s mean luminance gray period. Five temporal frequencies (1, 2, 4, 8, 15 Hz), eight different directions (separated by 45°), and one spatial frequency (0.04 cycles per degree) were used. Each grating condition was presented 15 times in random order.

The static gratings stimulus consisted of a full field sinusoidal grating at 80% contrast that was flashed for 250 ms, with no intervening gray period. Five spatial frequencies (0.02, 0.04, 0.08, 0.16, 0.32 cycles per degree), four phases (0, 0.25, 0.5, 0.75), and six orientations (separated by 30°) were used. Each grating condition was presented approximately 50 times in random order.

The natural scenes stimulus consisted of 118 natural images taken from the Berkeley Segmentation Dataset (*Martin et al., 2001*), the van Hateren Natural Image Dataset (*van Hateren and van der Schaaf, 1998*), and the McGill Calibrated Colour Image Database (*Olmos and Kingdom, 2004*). The images were presented in grayscale and were contrast normalized and resized to 1174 x 918 pixels. The images were presented in a random order for 0.25 s each, with no intervening gray period.

Two natural movie clips were taken from the opening scene of the movie Touch of Evil (*Welles, 1958*). Natural Movie One was a 30 s clips repeated 20 or 30 times (2 or 3 blocks of 10), while Natural Movie Three was a 120 s clip repeated 10 times (2 blocks of 5). All clips were contrast normalized and were presented in grayscale at 30 fps.

## Spikes-to-calcium forward model

All synthetic fluorescence traces were computed using MLSpike (*Deneux et al., 2016*) using the *third* model described in that paper. This version models the supra-linear behavior of the calcium fluorescence response function in the most physiological manner (out of the three models compared) by (1) explicitly accounting for cooperative binding between calcium and the indicator via the Hill equation and (2) including an explicit rise time, $\tau_{ON}$.

The model had seven free parameters: decay time ($\tau$), unitary response amplitude ($A$), noise level ($\sigma$), Hill exponent ($n$), $\Delta F/F$ rise time ($\tau_{ON}$), saturation ($\gamma$), and baseline calcium concentration ($c_0$). The last four parameters were fit on a ground truth dataset comprising 14 Emx1-Ai93 (from nine individual neurons across two mice) and 17 Cux2-Ai93 recordings (from 11 individual neurons across two mice), each between 120 s and 310 s in duration, with simultaneous cell-attached electrophysiology and two-photon imaging (noise-matched to the imaging dataset) (*Ledochowitsch et al., 2019*): $n = 2.42$, $\tau_{ON} = 0.0034$, $\gamma = 0.0021$, and $c_0 = 0.46$. Reasonable values for the first three parameters were established by applying the MLSpike autocalibration function to all neurons recorded in the imaging dataset, computing a histogram for each parameter, and choosing the value corresponding to the peak of the histogram, which yielded $\tau = 0.359$, $A = 0.021$, and $\sigma = 0.047$.

To convert spike times to synthetic fluorescence traces, MATLAB code publicly released by Deneux et al. (https://github.com/MLspike) was wrapped into a Python (v3.6.7) module via the MAT-LAB Library Compiler SDK, and run in parallel on a high-performance compute cluster.

## $\ell_0$-regularized event extraction

Prior to computing response metrics, the normalized fluorescence traces for both the experimental and synthetic imaging data were passed through an $\ell_0$ event detection algorithm that identified the onset time and magnitude of transients (*Jewell and Witten, 2018*; *Jewell et al., 2018*; *de Vries et al., 2020*), using a revised version of this algorithm available at github.com/jewellsean/FastLZero-SpikeInference. The half-time of the transient decay was assumed to be fixed at 315 ms. To avoid overfitting small-amplitude false-positive events to noise in the fluorescence trace, the $\ell_0$-regularization was adjusted for each neuron such that the smallest detected events were at least 200% of the respective noise floor (computed as the robust standard deviation of the noise via the `noise_std ()` Python function from the allensdk.brain_observatory.dff module) using an iterative algorithm. All subsequent analyses were performed on these events, rather than continuous fluorescence time series.

## Non-nonegative deconvolution (NND)

For the comparisons shown in *Figure 3*, we also extracted events via non-negative deconvolution (NND), using the Python implementation included in Suite2p (*Pachitariu et al., 2016b*). Prior to extracting events with NND, we upsampled the 30 Hz $\Delta F/F$ traces to 150 Hz using scipy.signal. resample_poly because in our hands NND performed substantially better on upsampled data (*Huang et al., 2021*). In another benchmarking paper (*Berens et al., 2018*), data were also upsampled to 100 Hz 'for ease of comparison'.

To filter out low-amplitude events, we first scaled the NND event amplitudes by the maximum value of the original $\Delta F/F$ trace for that neuron, which allowed us to define an event magnitude threshold as a function of the noise level detected in the $\Delta F/F$ trace. We then binned events in 100 ms intervals by defining the bin's magnitude as the sum over the magnitudes of all events that fell in that bin. We set the value of a bin to zero if its overall event magnitude was below a threshold value that was an integer multiple ($0 \leq n \leq 10$) of each neuron's robust standard deviation of the noise ($\sigma$).

## Visual response metrics

All response metrics were calculated from data stored in NWB 2.0 files (*Ruebel et al., 2019*) using Python code in a custom branch of the AllenSDK (github.com/jsiegle/AllenSDK/tree/ophys-ephys),

which relies heavily on NumPy (*van der Walt et al., 2011*), SciPy (SciPy 1.0 *SciPy 1.0 Contributors et al., 2020*), Matplotlib (*Hunter, 2007*), Pandas (*McKinney, 2010*), xarray (*Hoyer and Hamman, 2017*), and scikit-learn (*Pedregosa et al., 2012*) open-source libraries. For a given *imaging* stimulus presentation, the response magnitude for one neuron was defined as the summed amplitude of all of the events occurring between the beginning and end of the presentation. For a given *ephys* stimulus presentation, the response magnitude for one neuron was defined as the number of spikes occurring between the beginning and end of the presentation. Otherwise, the analysis code used for the two modalities was identical.

## Responsiveness

To determine whether a neuron was responsive to a given stimulus type, the neuron's response to its preferred condition was compared to a distribution of its activity during the nearest epoch of mean-luminance gray screen (the 'spontaneous' interval). This distribution was assembled by randomly selecting 1000 intervals with the same duration of each presentation for that stimulus type (drifting gratings = 2 s, static gratings = 0.25 s, natural scenes = 0.25 s, natural movies = 1/30 s). The preferred condition is the stimulus condition (e.g. a drifting grating with a particular direction and temporal frequency) that elicited the largest mean response. The *response reliability* was defined as the percentage of preferred condition trials with a response magnitude larger than 95% of spontaneous intervals. A neuron was deemed *responsive* to a particular stimulus type if its response reliability was greater than 25%. Selectivity and preference metrics were only analyzed for responsive neurons.

## Selectivity

The selectivity of a neuron's responses within a stimulus type was measured using a *lifetime sparseness* metric (*Vinje and Gallant, 2000*). Lifetime sparseness is defined as:

$$\frac{1 - \frac{1}{n} \cdot \left(\sum_{i=1}^{n} r_i\right)^2 \cdot \left(\sum_{i=1}^{n} r_i^2\right)^{-1}}{1 - \frac{1}{n}}$$

where $n$ is the total number of conditions, and $r_i$ represents the response magnitude for condition $i$. If a neuron has a non-zero response to only one condition (maximally selective response), its lifetime sparseness will be 1. If a neuron responds equally to all conditions (no selectivity), its lifetime sparseness will be 0. Importantly, lifetime sparseness is a nonparametric statistic that considers a neuron's selectivity across all possible stimulus conditions within a stimulus type, rather than conditions that vary only one parameter (e.g. orientation selectivity). For that reason, it is applicable to any stimulus type.

## Preference

For all stimulus types, the preferred condition was defined as the condition (or frame, in the case of natural movies) that elicited the largest mean response across all presentations. For drifting gratings, the preferred temporal frequency was defined as the temporal frequency that elicited the largest mean response (averaged across directions). For static gratings, the preferred spatial frequency was defined as the spatial frequency that elicited the largest mean response (averaged across orientations and phases).

## Matching layer distributions

Neurons in the imaging dataset were assigned to layers based on the depth of the imaging plane (<200 μm = L2/3, 200–325 μm = L4, 325–500 μm = L5, >500 μm = L6), or the mouse Cre line (Nr5a1-Cre and Scnn1a-Tg3-Cre neurons were always considered to be L4). Neurons in the ephys dataset were assigned to layers after mapping their position to the Common Coordinate Framework version 3 (*Wang et al., 2020*). CCFv3 coordinates were used as indices into the template volume in order to extract layer labels for each cortical unit (see *Siegle et al., 2021* for details of the mapping procedure).

To test for an effect of laminar sampling bias, L6 neurons were first removed from both datasets. Next, since the ephys dataset always had the highest fraction of neurons L5, neurons from L2/3 and L4 of the imaging dataset were randomly sub-sampled to match the relative fraction of ephys

neurons from those layers. The final resampled layer distributions are shown in *Figure 2—figure supplement 2B*.

## Burst metrics

Bursts were detected using the *LogISI* method (*Pasquale et al., 2010*). Peaks in the histogram of the log-adjusted inter-spike intervals (ISI) were identified, and the largest peak corresponding to an ISI of less than 50 ms was set as the intra-burst peak. In the absence of such a peak, no bursts were found. Minima between intra-burst peak and subsequent peaks were found, and a void parameter, representing peak separability, was calculated for each minimum. The ISI value for the first minimum where the void parameter exceeds a default threshold of 0.7 was used as the *maxISI*-cutoff for burst detection. Bursts were then defined as a series of >three spikes with ISIs less than maxISI. If no cutoff was found, or if *maxISI* > 50 ms, burst cores were found with <50-ms ISI, and any spikes within *maxISI* of burst edges were included.

R code provided with a comparative review of bursting methods (*Cotterill et al., 2016*; https://github.com/ellesec/burstanalysis) was wrapped into Python (v.3.6.7) using the rpy2 interface (https://rpy2.github.io), and run in parallel on a high-performance compute cluster.

## Statistical comparisons

Jensen–Shannon distance was used to quantify the disparity between the distributions of metrics from imaging and ephys. This is the square root of the Jensen–Shannon divergence, also known as the total divergence to the mean, which, while derived from the Kullback–Leibler divergence, has the advantage of being symmetric and always has a finite value. The Jensen–Shannon distance constitutes a true mathematical metric that satisfies the triangle inequality. (*Lin, 1991*). We used the implementation from the SciPy library (SciPy 1.0 *SciPy 1.0 Contributors et al., 2020*; scipy.spatial.jensenshannon). For selectivity and responsiveness metrics, Jensen–Shannon distance was calculated between histograms with 10 equal-sized bins between 0 and 1. For preference metrics, Jensen–Shannon distance was calculated between the preferred condition histograms, with unit spacing between the conditions.

To compute p values for Jensen–Shannon distances, we used a bootstrap procedure that randomly sub-sampled metric values from one modality, and calculated the distance between these intra-modal distributions. We repeated this procedure 1000 times, in order to calculate the probability that the true inter-modality distance would be less than the distance between the distributions of two non-overlapping intra-modality samples.

The Pearson correlation coefficient (scipy.stats.pearsonr) was used to quantify the correlation between two variables. The Mann–Whitney *U* test (scipy.stats.ranksums) was used to test for differences in running speed or running fraction between the imaging and ephys datasets.

## Clustering of response reliabilities

We performed a clustering analysis using the response reliabilities by stimulus for each neuron (defined as the percentage of significant trials to the neuron's preferred stimulus condition), across drifting gratings, static gratings, natural scenes, and natural movies. We combined the reliabilities for natural movies by taking the maximum reliability over Natural Movie One and Natural Movie Three. This resulted in a set of four reliabilities for each neuron (for drifting gratings, static gratings, natural movies, and natural scenes).

We performed a Gaussian Mixture Model clustering on these reliabilities for cluster numbers from 1 to 50, using the average Bayesian Information Criterion (*Schwarz, 1978*) on held-out data with four-fold cross validation to select the optimal number of clusters. Once the optimal model was selected, we labeled each cluster according to its profile of responsiveness (i.e. the average reliability across all neurons in the cluster to drifting gratings, static gratings, etc.), defining these profiles as 'classes'. For each neuron, we predicted its cluster membership using the optimal model, and then the class membership using a predefined responsiveness threshold. We repeated this process 100 times to estimate the robustness of the clustering and derive uncertainties for the number of cells belonging to each class.

## Acknowledgements

We thank the Allen Institute founder, Paul G Allen, for his vision, encouragement, and support. We thank the MindScope Scientific Advisory Council for providing feedback on our initial results. We thank the TissueCyte team, including Nadezhda Dotson, Mike Taormina, and Anh Ho, for histology and imaging associated with *Figure 3A*. We thank Adam Charles, Jeff Gauthier, and David Tank for helpful discussions on data processing methods. We thank Nicholas Cain for developing the computational infrastructure that made this project feasible, as well as for his enthusiastic analysis of pilot data in the early stages of this project. We thank Corbett Bennett and Jun Zhuang for their feedback on the manuscript. MAB received funding from NIH award 1R01EB026908-01.

## Additional information

### Funding

| Funder | Grant reference number | Author |
| --- | --- | --- |
| Allen Institute | | Joshua H Siegle |
| | | Peter Ledochowitsch |
| | | Xiaoxuan Jia |
| | | Daniel J Millman |
| | | Gabriel K Ocker |
| | | Shiella Caldejon |
| | | Linzy Casal |
| | | Andy Cho |
| | | Daniel J Denman |
| | | Séverine Durand |
| | | Peter A Groblewski |
| | | Gregg Heller |
| | | India Kato |
| | | Sara Kivikas |
| | | Jérôme Lecoq |
| | | Chelsea Nayan |
| | | Kiet Ngo |
| | | Philip R Nicovich |
| | | Kat North |
| | | Tamina K Ramirez |
| | | Jackie Swapp |
| | | Xana Waughman |
| | | Ali Williford |
| | | Shawn R Olsen |
| | | Christof Koch |
| | | Michael A Buice |
| | | Saskia EJ de Vries |
| NIH | 1R01EB026908-01 | Michael A Buice |

The funders had no role in study design, data collection and interpretation, or the decision to submit the work for publication.

### Author contributions

Joshua H Siegle, Peter Ledochowitsch, Conceptualization, Formal analysis, Writing - original draft, Writing - review and editing; Xiaoxuan Jia, Daniel J Millman, Gabriel K Ocker, Daniel J Denman, Christof Koch, Conceptualization, Writing - review and editing; Shiella Caldejon, Andy Cho, Gregg Heller, India Kato, Sara Kivikas, Chelsea Nayan, Kiet Ngo, Kat North, Tamina K Ramirez, Jackie Swapp, Xana Waughman, Investigation; Linzy Casal, Project administration; Séverine Durand, Supervision, Investigation; Peter A Groblewski, Jérôme Lecoq, Philip R Nicovich, Ali Williford, Supervision; Shawn R Olsen, Supervision, Writing - review and editing; Michael A Buice, Conceptualization, Formal analysis, Supervision, Writing - review and editing; Saskia EJ de Vries, Conceptualization, Formal analysis, Supervision, Writing - original draft, Writing - review and editing

### Author ORCIDs

Joshua H Siegle ⓘ https://orcid.org/0000-0002-7736-4844
Peter Ledochowitsch ⓘ https://orcid.org/0000-0003-0835-3444

Daniel J Millman [ID] http://orcid.org/0000-0002-6255-6085
Gabriel K Ocker [ID] https://orcid.org/0000-0001-9627-9576
Peter A Groblewski [ID] http://orcid.org/0000-0002-8415-1118
Shawn R Olsen [ID] http://orcid.org/0000-0002-9568-7057
Michael A Buice [ID] http://orcid.org/0000-0002-2196-1498
Saskia EJ de Vries [ID] https://orcid.org/0000-0002-3704-3499

### Ethics

Animal experimentation: All animal procedures were approved by the Institutional Animal Care and Use Committee (IACUC) at the Allen Institute in compliance with NIH guidelines.

### Decision letter and Author response

Decision letter https://doi.org/10.7554/eLife.69068.sa1
Author response https://doi.org/10.7554/eLife.69068.sa2

## Additional files

### Supplementary files

• Transparent reporting form

### Data availability

Code and intermediate data files required to generate all manuscript figures are available through GitHub (https://github.com/AllenInstitute/ophys_ephys_comparison_paper). NWB files for the imaging and ephys sessions (excluding the ephys experiments in GCaMP6 mice) are available via the AllenSDK (https://github.com/AllenInstitute/AllenSDK, copy archived at https://archive.softwareheritage.org/swh:1:rev:b29058f4c136f906123b3f93cd184edb9158b4e4), and also as an AmazonWeb Services Public Data Set (https://registry.opendata.aws/allen-brain-observatory/). NWB files for the ephys experiments in GCaMP6 mice are available through Zenodo (https://https://zenodo.org/record/5090747).

The following dataset was generated:

| Author(s) | Year | Dataset title | Dataset URL | Database and Identifier |
|---|---|---|---|---|
| Siegle JH, Ledochowitsch P, Jia X, Millman DJ, Ocker GK, Caldejon S, Casal L, Cho A, Denman DJ, Durand S, Groblewski PA, Heller G, Kato I, Kivikas S, Lecoq J, Nayan C, Ngo K, Nicovich PR, North K, Ramirez TK, Swapp J, Waughman X, Williford A, Olsen SR, Koch C, Buice MA, de Vries SE | 2021 | Neuropixels recordings from GCaMP6f+ mice | https://zenodo.org/record/5090747 | Zenodo, 10.5281/zenodo.5090747 |

The following previously published datasets were used:

| Author(s) | Year | Dataset title | Dataset URL | Database and Identifier |
|---|---|---|---|---|
| Allen Institute for Brain Science | 2016 | Allen Brain Observatory Visual Coding | http://observatory.brainmap.org/visualcoding/ | Allen Brain Atlas, observatory.brainmap.org/visualcoding/ |

Allen Institute for Brain Science 2019 Allen Brain Observatory Neuropixels https://portal.brain-map.org/explore/circuits/visual-coding-neuropixels Allen Brain Map, portal.brain-map.org/explore/circuits/visual-coding-neuropixels

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
