## [Decision Letter]

**Acceptance summary:**

This paper addresses the important topic of comparing the data obtained from standard electrophysiology recordings to calcium imaging in a controlled experimental set up [it does ephys and 2p in different mice, perhaps not ideal but very useful for real comparisons of different datasets]. Previously, most comparisons have been performed at the level of single cells with simultaneous ephys and imaging, and these comparisons have not addressed the topics discussed here, which are highly relevant to the field. The discussion and analysis of pros and cons to each method are fair and valuable, in particular as the shortcomings of electrophysiology are sometimes not discussed in the literature.

**Decision letter after peer review:**

[Editors’ note: the authors submitted for reconsideration following the decision after peer review. What follows is the decision letter after the first round of review.]

Thank you for submitting your work entitled "Reconciling functional differences in populations of neurons recorded with two-photon imaging and electrophysiology" for consideration by *eLife*. Your article has been reviewed by 3 peer reviewers, and the evaluation has been overseen by a Reviewing/Senior Editor. The reviewers have opted to remain anonymous.

Our decision has been reached after consultation between the reviewers. Based on these discussions and the individual reviews below, we regret to inform you that your work will not be considered further for publication in *eLife*.

While all the reviewers saw great merit in the approach, there are several flaws large and small that require a thorough reworking of the paper and significant new analyses that could substantially change the main conclusions. Such a thoroughly reworked paper taking into account all the reviewer comments could be submitted as a new submission to *eLife*.

*Reviewer #1:*

In this manuscript, Siegle, Ledochowitsch et al. compare two large datasets of neurons recorded by electrophysiology and two-photon calcium imaging respectively. They start by showing that there are substantial differences in the responsiveness and selectivity of the neurons, but those differences mostly go away after taking into account the details of these methods. This work is very useful information for anyone using ephys or ophys to record neurons from brains, which includes a *lot* of labs. It can become a priceless reference for everyone worried about what their data actually represents, and there is a lot of concern going around these days, especially as the recording methods are being scaled up very rapidly.

The paper and the analysis are very well written and convey all the information we need. I have almost no minor or major comments, but I do have one very serious concern. The paper is not really comparing ephys to ophys, but comparing ephys to L0-deconvolution of calcium imaging data. While I believe deconvoluton is absolutely crucial for interpreting 2p data correctly, the method chosen by the authors – L0 deconvolution – is problematic because it *will* result in exactly the kind of selectivity and responsiveness effects reported here. This is not merely a possibility, but a guaranteed consequence of L0-deconvolution because this method by design returns a very large fraction of zeros, effectively only picking up the "top of the iceberg" from the calcium traces.

The solution to this problem is clear: adjust the regularization constant in the L0 deconvolution all the way down to 0 and see how your results change. The Jewell et al. method will probably stop working correctly before you get to 0, in which case please use un-penalized OASIS, which is deconvolution with no regularization. Suite2p provides wrappers to do this with everything turned off (no parameter estimation and no regularization), and there are papers out there showing that the optimal value of the regularization constant is actually 0 for some ground truth data.

I would be extremely surprised if the results don't change substantially, and I am very curious to know exactly how they change.

*Reviewer #2:*

This paper addresses the important topic of comparing the data obtained from standard electrophysiology recordings to calcium imaging in a controlled experimental set up. Previously, most comparisons have been performed at the level of single cells with simultaneous ephys and imaging, and these comparisons have not addressed the topics discussed here, which are highly relevant to the field. Overall the paper and results are well presented. The conclusions and interpretations follow from the data presented. The discussion and analysis of pros and cons to each method are fair and valuable, in particular as the shortcomings of electrophysiology are sometimes not discussed in the literature. I therefore think this paper will be a valuable addition to the field. However, there are some shortcomings that need to be noted clearly and some potential extensions that could further enhance the impact of the work.

1. The major shortcoming of the paper is that it is focused on only a single labeling strategy for calcium imaging, namely using transgenic mice with the GCaMP6f indicator. The labeling strategy and indicator likely have very major impacts on the data obtained from calcium imaging. Specifically, expression levels can vary depending on the labeling approach, such as between viral methods and transgenic methods, which can affect the signal-to-noise ratio and the ability to detect single action potentials. Also, different indicators will have different detection thresholds. Here GCaMP6f was used, which is relatively insensitive, whereas a choice of GCaMP6s would have resulted in more detection of single action potentials. It is well known among experimenters in the field that viral labeling generally gives substantially higher signal-to-noise ratio compared to transgenic methods, including the transgenic methods used in this paper. This can be seen in some of the single-spike detection rates shown in the original GCaMP6 paper compared to subsequent efforts from the Allen Institute and others using transgenic mice. The same is true for GCaMP6s vs. GCaMP6f as was shown clearly in the original GCaMP6 paper. It is therefore the case that the differences between ephys and imaging could be different if viral methods were used and if GCaMP6s or GCaMP7 variants were used for the experiments. This is really important and currently the paper does not put emphasis on this point. The danger is that readers will think these results generalize to all calcium imaging studies, which I think is unlikely to be true, at least in the details, but perhaps in the general findings too. Obviously it is too much work to repeat all the experiments with a different indicator and/or labeling strategy. But, this is a critical point that must be addressed and presented very clearly. The authors must mention in the abstract that they used GCaMP6f and transgenic mice because these details are critical to the specifics of their findings. Furthermore, they should provide a detailed section in the discussion that points out that the results could vary substantially if different labeling methods and/or indicators are used. These issues raise another level of details regarding comparing studies, and it is critical that the authors discuss them thoroughly.

2. I very much liked the forward model to transform spikes to calcium events. I think it would also be useful to try deconvolution to transform calcium to spikes as well. I agree with the authors that this is an under-constrained problem without an ideal solution. However, there are common methods in the field that are frequently applied in calcium imaging papers. I suggest the authors use a common deconvolution approach and compare how this impacts the comparison between ephys and calcium imaging. I suggest this because most calcium imaging studies now analyze deconvolved data instead of the calcium events.

3. One of the very important findings is that the distribution of tuning preferences look to be similar between methods, for example for temporal frequency in Figure 2G. I think this point should be emphasized even more to point out the things that are similar between methods. I would also like to see these same distributions for other tuning parameters, such as spatial frequency, orientation, direction, and so on. It was not clear why the authors only showed temporal frequency.

4. It would be helpful to see additional metrics for selectivity beyond lifetime sparseness. I like the lifetime sparseness metric, but there are others that are more commonly used in the field and might help the reader to see. As one possibility, it might be helpful to see an orientation selectivity index as a comparison between ephys and calcium imaging data. I understand that this is not a general metric, but it is common in the field and thus potentially helpful.

5. The responsiveness measure seems reasonable, but it is based on a seemingly arbitrary threshold of 25% of trials with responses above 95% of the baseline period. What happens if this 25% threshold is varied? In addition, other measures of response reliability might be helpful. For example, trial-to-trial variability might be an interesting and important parameter to quantify, such as using the coefficient of variation or other metrics.

6. My understanding is that the calcium imaging data were segmented into cells using morphology based methods. This is not the current standard in the field. Most papers now use correlations in the fluorescence timeseries to help with segmentation. Is the segmentation obtained by the authors consistent with the segmentation obtained with more standard methods in the field, such as the cells obtained with Suite2P or CNMF? It might also be helpful for the authors to discuss how cell segmentation with calcium imaging, including neuropil contamination/subtraction, could impact the metrics they studied. The analysis of potential spike sorting artifacts or biases was informative, and something analogous could be, at the least, discussed for calcium imaging segmentation.

7. I do not like the introduction of the term 'ophys'. To me, this just adds another term to the field and likely results in more confusion rather than sticking with normal nomenclature in the field. I do not see what it adds over referring to the data as calcium imaging or two photon calcium imaging. If anything 'ophys' seems less precise because it could include voltage imaging too.

*Reviewer #3:*

This paper makes a step towards addressing a need in the community: understanding if measurements made with electrophysiology (ephys) and 2-photon imaging (2p) give the same result, and if not, in which way they differ and why. The best way to compare the two would be to do ephys and 2p at the same time (e.g. Wei,.… Druckmann, bioRxiv 2019). Next best would be to do them at different times but in the same neurons (which could be done here, because many of the ephys mice express GCaMP6f). This paper uses the third best approach: it does ephys and 2p in different mice. Not ideal but a useful exercise given the remarkably vast datasets.

The paper however contains some flawed analyses and meanders away from the main focus. It could be refocused e.g. by keeping the top half of Figure 1; a revised Figure 2; Figure 4; the top half of Figure 5, and a panel similar to Figure 4E, showing that once biases in cell selection are taken care of, the fraction of responsive neurons to different stimuli becomes the same. The current Figures 2 and 7 show analyses that are simply not ok, as explained below.

(1) The paper starts by making a flawed comparison between 2p and ephys data (Figure 2). One can't directly compare the output of ephys, i.e. spike trains that are sequences of 1s and 0s, with the deconvolved output of 2p, i.e. firing rates that range over 4 orders of magnitude. To compare those two measures of activity they must have the same units: either compute firing rates with common bin size (by binning the spike trains and resampling the deconvolved firing rates) or convolve the spike trains with a filter to match the GCaMP timecourse (as in Figure 4).

(2) Based on Figure 2, the paper finds two discrepancies: the 2p dataset has neurons with higher stimulus selectivity (Figure 2H) but a lower fraction of responsive neurons (Figure 2F). The first discrepancy disappears once the spikes measured with ephys are convolved with a reasonable filter that models the impulse response of GCaMP (Figure 4I). The second discrepancy remains (Figure 4E) but it disappears (does it? this is not fully shown) if one considers that the ephys data is biased towards neurons that fire more.

(3) The analysis of cell selection bias takes 3 long figures (Figure 5,6, and 7) without delivering the desired result: we need to see a figure with the same format as Figure 1F and Figure 4I, showing a match between the two techniques -- if such a match can be obtained.

(4) To address sampling bias introduced by the 2p processing pipeline, the paper explores the effect of setting a threshold at a given firing rate (p 13). However, the sampling bias may be more closely related to the procedures used to identify neurons in the 2p images. It seems that ROIs here are based on morphological properties without further manual curation (de Vries et al., 2020). This wide net will increase the total number of neurons, and thus likely decrease the fraction of responsive neurons, as pointed out by the same group (Mesa et al., bioRxiv). Subsampling using event rate reduces rather than increases overall selectivity, but in the Mesa et al. preprint, subsampling using a higher threshold for coefficient of variation (CV) increases selectivity with fewer neurons. So, at the very least, consider subsampling using CV to see whether this increases similarity between recording methods.

(5) Another reason for high sparseness/selectivity in the 2p data could be the deconvolution algorithm. Indeed L0 deconvolution, and L0 regularization in general, are intended to sparsify (e.g. as described in Pachitariu et al., 2018 J Neurosci). This could be dealt with by re-analyzing the dataset with different deconvolution algorithms, or vary the parameter λ governing the L0/L1 trade-off. Or perhaps better, one could make synthetic 2p data by convolving the ephys data and then deconvolve it to estimate firing rates. If they become sparser, there is a problem.

(6) The statistical test of Jensen-Shannon distance seems to find significant differences also where there barely are any. In some analyses (e.g. Figure 4I) the distributions look highly similar, yet are still judged to be statistically distinct. Is there an instance (e.g. within the same modality) where the Jensen-Shannon distance would ever be small enough to fail to reject the null? For example, by taking two sets of ephys data obtained from the same population in two different days.

(7) Figure 7 shows an unconvincing attempt at clustering neurons. It rests on a clustering process that seems dubious (t-SNE is geared to show data as clusters) and would require a whole paper to be convincing. It hinges on the existence and robustness of clusters of cells based on responses to a variety of stimuli. If there are clusters of that kind, that's a major discovery and it should get its own paper. But showing it convincingly would likely require long recordings (e.g. across days). It is not the kind of thing one shows at the end of a paper on a different topic. What we want at the end of this paper is to know whether the correction for selection bias explains the discrepancy shown in the simple analysis of Figure 4.

(8) The separate analysis of different visual areas adds complexity without delivering benefits. There is little reason to think that the relationship between spikes and calcium is different in neighboring cortical areas. If it is, then it's an interesting finding, but one that will require substantial burden of proof (as in simultaneous recordings). Otherwise it seems more logical to pool the neurons, greatly simplifying the paper. (This said, it does seem somewhat useful to separate the visual areas in Figure 4G, as they do have some differences in selectivity).

(9) The paper would be much stronger if it included an analysis of GCaMP6s data in addition to GCaMP6f. Are the same findings made with those mice?

(10) Are there consistent differences in reliability of the responses (repeatability across trials of same stimuli) in the two measures, and if so, does the convolution and selection biases explain these differences.

[Editors’ note: further revisions were suggested prior to acceptance, as described below.]

Thank you for resubmitting your work entitled "Reconciling functional differences in populations of neurons recorded with two-photon imaging and electrophysiology" for further consideration by *eLife*. Your revised article has been evaluated by Ronald Calabrese as the Senior and Reviewing Editor.

The manuscript has been improved but there are some remaining issues that need to be addressed, as outlined below:

The reviewers found the paper much improved but there is still some concern that the message is somewhat muddled and that readers are not being given the guidance they need to treat their data appropriately. In particular, please make sure readers come away with guidance on what to do with their data; please discuss this more and clarify this issue throughout the text. Recommendations for the Authors of Reviewer #1 will be helpful.

*Reviewer #1:*

My earlier review had suggested ways to make this paper briefer and more incisive so that it would have a clearer message. To summarize my opinion, the message would have been clearer if the paper started by listing the differences between the two methods (using an apples-to-apples comparison from the get-go) and then introduced the forward model and step by step indicated what one needs to do to make the differences go away. I would not introduce in the last figure a new result/analysis (the clusters of cell types) and use that one to illustrate the success of the forward model. I would drop that figure because it's a dubious (potentially interesting) result that would need its own paper to be convincing. By the time we arrive at that figure we should have established how we will judge a model's capabilities, rather than throw more data at the reader. This said, there are certainly many ways to write a paper, and it should be ok to ignore one reviewer's opinion on how to arrange an argument.

My main suggestion is to give the paper another look and try to convey a clearer message. For instance, take this sentence in abstract: "However, the forward model could not reconcile differences in responsiveness without sub-selecting neurons based on event rate or level of signal contamination." If one took out the double negative (not/without) one would end up with a positive result: the forward model can indeed reconcile those differences, provided that one focuses on neurons with low contamination and sets a threshold for event rate. Writing things this way would actually honor the goal stated in the title: "Reconciling functional differences…". Overall, it would be good if readers could come out with a clearer understanding of what they should do to their 2p data to make it more similar to ephys data, and what aspects they can trust and what aspects they cannot.

*Reviewer #2:*

Looks great to me, thanks for doing the extra analyses from Figure 3!

*Reviewer #3:*

I support publication of this paper and think it will be a nice addition to the field and of interest to a large number of readers. The paper is clearly written and presented. The addition of Figure 3 and new discussion points is helpful. I do not have remaining concerns or comments.

---

## [Author Response]

Reviewer #1:In this manuscript, Siegle, Ledochowitsch et al. compare two large datasets of neurons recorded by electrophysiology and two-photon calcium imaging respectively. They start by showing that there are substantial differences in the responsiveness and selectivity of the neurons, but those differences mostly go away after taking into account the details of these methods. This work is very useful information for anyone using ephys or ophys to record neurons from brains, which includes a lot of labs. It can become a priceless reference for everyone worried about what their data actually represents, and there is a lot of concern going around these days, especially as the recording methods are being scaled up very rapidly.The paper and the analysis are very well written and convey all the information we need. I have almost no minor or major comments, but I do have one very serious concern. The paper is not really comparing ephys to ophys, but comparing ephys to L0-deconvolution of calcium imaging data. While I believe deconvoluton is absolutely crucial for interpreting 2p data correctly, the method chosen by the authors – L0 deconvolution --- is problematic because it will result in exactly the kind of selectivity and responsiveness effects reported here. This is not merely a possibility, but a guaranteed consequence of L0-deconvolution because this method by design returns a very large fraction of zeros, effectively only picking up the "top of the iceberg" from the calcium traces.The solution to this problem is clear: adjust the regularization constant in the L0 deconvolution all the way down to 0 and see how your results change. The Jewell et al. method will probably stop working correctly before you get to 0, in which case please use un-penalized OASIS, which is deconvolution with no regularization. Suite2p provides wrappers to do this with everything turned off (no parameter estimation and no regularization), and there are papers out there showing that the optimal value of the regularization constant is actually 0 for some ground truth data.I would be extremely surprised if the results don't change substantially, and I am very curious to know exactly how they change.

The reviewer brings up a crucial point: the approach to extracting dF/F events *can* have an impact on functional metrics. Following the reviewer’s advice, we have carried out a novel comparison of event detection methods that adds a valuable new dimension to the manuscript. Our overall conclusion – that selectivity and responsiveness can be biased by the presence/absence of low-amplitude events – represents an important contribution to the literature, which to our knowledge has not been considered in detail elsewhere.

We had previously selected the Jewell and Witten “exact L_0_” deconvolution method after extensive characterization of its performance on our ground truth dataset. We found that it performs almost as well as a more complex method, MLSpike, and is virtually indistinguishable from non-negative deconvolution in terms of its ability to predict ground truth firing rates. The main difference between exact L0 and NND – as pointed out by the reviewer – is that exact L0 returns far fewer events, due to the imposed sparsity constraint. Both methods reliably convert large changes in dF/F to events, but unpenalized NND additionally detects a larger number of small-amplitude events. Because these have a negligible impact on the correlation between the event time series and ground truth firing rate (since a tiny event is still a good estimate of 0 Hz spiking), we did not originally consider the impact these events could have on functional metrics. However, we have now demonstrated that they do change the apparent responsiveness and selectivity, to the point where it would be impossible to accurately compare results across imaging studies without taking the event detection methods into consideration. This point was made to some degree in the bioRxiv manuscript of Evans et al. (2019), which compared the scientific conclusions that resulted from applying various event detection algorithms to calcium responses measured in somatosensory cortex. We now explore these impacts in a more systematic way, and link them to a particular feature of deconvolution, the background rate of low-amplitude events.

We present these results in a new figure in the main text, Figure 3. When using unpenalized NND to extract events, we find that population selectivity levels are *lower* than we found with exact L0 (Figure 3F), while the responsive fraction of neurons is *higher* (Figure 3G). While at first glance this suggests a better match to the original ephys data, we find that these results are especially sensitive to the presence of low-amplitude events, which are quite prevalent in unpenalized NND. As we say in the manuscript, “By imposing the same noise-based threshold on the minimum event size that we originally computed to determine optimal regularization strength (essentially performing a post-hoc regularization), we are able to reconcile the results obtained with NND with those obtained via exact L0” (p. 10). We show that removing these events results in no change in the correlation with ground truth firing rate (3B), but a substantial reduction in “background” response magnitude (3C) as well as the rate of false positives (3D). These latter two factors have a big impact on selectivity and responsiveness – since events always have a positive magnitude, those detected near the noise floor cannot be canceled out by an equal fraction of negative-going events, as would occur when analyzing the DF/F trace directly.

Prior work validating event extraction methods has focused almost exclusively on maximizing the correlation between ground truth firing rate and total event amplitude within some time bin (but see Pachitariu et al., 2018 *J Neurosci* for a counterexample where response reliability was optimized instead). Here, we show that using correlation alone (which only quantifies a method’s ability to predict instantaneous firing rate) does not capture features of event extraction algorithms that have a clear impact on functional metrics (which average over many trials, and are thus quite sensitive to changes in background rate).

By modifying event extraction parameters in a way that affects the false positive rate of event detection, we can indeed increase the similarity between some metrics distributions across recording modalities. However, we caution against using this approach for the purposes of reconciliation across modalities because there is no unique parameter set that jointly reconciles responsiveness and selectivity, suggesting that this cannot be the whole story. We hesitate to treat the ephys data as ground truth in either case, since we know that spike trains can suffer from significant contamination.

We are grateful for the reviewer’s comments that inspired us to think more deeply about the effect of event extraction on the metrics we consider, culminating in the inclusion of the new analysis presented in Figure 3. It provides an essential complement to the revised manuscript’s Figure 5, in which we apply the same event extraction method on the real and simulated calcium traces, allowing us to account for the sparsifying effects of this analysis step.

Reviewer #2:This paper addresses the important topic of comparing the data obtained from standard electrophysiology recordings to calcium imaging in a controlled experimental set up. Previously, most comparisons have been performed at the level of single cells with simultaneous ephys and imaging, and these comparisons have not addressed the topics discussed here, which are highly relevant to the field. Overall the paper and results are well presented. The conclusions and interpretations follow from the data presented. The discussion and analysis of pros and cons to each method are fair and valuable, in particular as the shortcomings of electrophysiology are sometimes not discussed in the literature. I therefore think this paper will be a valuable addition to the field. However, there are some shortcomings that need to be noted clearly and some potential extensions that could further enhance the impact of the work.1. The major shortcoming of the paper is that it is focused on only a single labeling strategy for calcium imaging, namely using transgenic mice with the GCaMP6f indicator. The labeling strategy and indicator likely have very major impacts on the data obtained from calcium imaging. Specifically, expression levels can vary depending on the labeling approach, such as between viral methods and transgenic methods, which can affect the signal-to-noise ratio and the ability to detect single action potentials. Also, different indicators will have different detection thresholds. Here GCaMP6f was used, which is relatively insensitive, whereas a choice of GCaMP6s would have resulted in more detection of single action potentials. It is well known among experimenters in the field that viral labeling generally gives substantially higher signal-to-noise ratio compared to transgenic methods, including the transgenic methods used in this paper. This can be seen in some of the single-spike detection rates shown in the original GCaMP6 paper compared to subsequent efforts from the Allen Institute and others using transgenic mice. The same is true for GCaMP6s vs. GCaMP6f as was shown clearly in the original GCaMP6 paper. It is therefore the case that the differences between ephys and imaging could be different if viral methods were used and if GCaMP6s or GCaMP7 variants were used for the experiments. This is really important and currently the paper does not put emphasis on this point. The danger is that readers will think these results generalize to all calcium imaging studies, which I think is unlikely to be true, at least in the details, but perhaps in the general findings too. Obviously it is too much work to repeat all the experiments with a different indicator and/or labeling strategy. But, this is a critical point that must be addressed and presented very clearly. The authors must mention in the abstract that they used GCaMP6f and transgenic mice because these details are critical to the specifics of their findings. Furthermore, they should provide a detailed section in the discussion that points out that the results could vary substantially if different labeling methods and/or indicators are used. These issues raise another level of details regarding comparing studies, and it is critical that the authors discuss them thoroughly.

Following the reviewer’s suggestion, we have updated the abstract to emphasize that we are using transgenic mice: “Here, we compare evoked responses in visual cortex recorded in awake mice under highly standardized conditions using either imaging of genetically expressed GCaMP6f or electrophysiology with implanted silicon probes.”

While more experimental data is needed to verify that our results generalize to other calcium indicators or expression methods, we have reason to believe that they will. In our analysis of the forward model parameter sweep (which now appears in Figure S5), we see very little effect of fluorescence decay time on functional metrics for the same cells. This indicates that, for the same underlying physiology, we do not see differences when a GCaMP6s-like forward model is used. We have added a paragraph to the discussion that considers this point in more detail (p. 23), which is reproduced here:

“One potential limitation of our approach is that we have only imaged the activity of transgenically expressed calcium indicators, rather than indicators expressed using a viral approach. […] The full range of this sweep includes decay times that are consistent with GCaMP6s, and event amplitudes that are consistent with viral expression.”

2. I very much liked the forward model to transform spikes to calcium events. I think it would also be useful to try deconvolution to transform calcium to spikes as well. I agree with the authors that this is an under-constrained problem without an ideal solution. However, there are common methods in the field that are frequently applied in calcium imaging papers. I suggest the authors use a common deconvolution approach and compare how this impacts the comparison between ephys and calcium imaging. I suggest this because most calcium imaging studies now analyze deconvolved data instead of the calcium events.

Our approach to extracting events from the calcium fluorescence time series performs a similar function as other commonly used deconvolution methods: it transforms a continuous DF/F trace into a series of events with a time and an amplitude. We do not believe these events can be assumed to correspond to spikes, and are often only weakly correlated with the true underlying spike rate. Our own work (Huang, Ledochowitsch et al., 2021) and that of others (see e.g., Rupprecht et al., 2021) has shown that reliably detecting single action potentials in population recordings that use GCaMP6 is effectively impossible.

In order to address this point, Figure 2 now includes a detailed illustration of how response magnitudes are calculated for both ephys and imaging. We have also added a new main text figure that compares our original approach to event extraction, with one based on non-negative deconvolution (Figure 3). We find that the rate of low-amplitude (often false-positive) events has a big impact on functional metrics, an important new finding that, to the best of our knowledge, has not been previously addressed in the literature.

3. One of the very important findings is that the distribution of tuning preferences look to be similar between methods, for example for temporal frequency in Figure 2G. I think this point should be emphasized even more to point out the things that are similar between methods. I would also like to see these same distributions for other tuning parameters, such as spatial frequency, orientation, direction, and so on. It was not clear why the authors only showed temporal frequency.

We now include a new Supplementary Figure 1 comparing tuning preferences for the metrics listed by the reviewer. In general, there is good correspondence between the two modalities. Minor discrepancies include bias toward 0º drifting gratings in ophys, and bias toward lower spatial frequencies in imaging.

4. It would be helpful to see additional metrics for selectivity beyond lifetime sparseness. I like the lifetime sparseness metric, but there are others that are more commonly used in the field and might help the reader to see. As one possibility, it might be helpful to see an orientation selectivity index as a comparison between ephys and calcium imaging data. I understand that this is not a general metric, but it is common in the field and thus potentially helpful.

We now show the distributions of orientation selectivity and direction selectivity in panels D and E of Supplementary Figure 1, which show the same trend as lifetime sparseness. In general, we prefer to use lifetime sparseness for our primary analysis, because this metric can be applied to all stimulus types. However, it is also helpful to present the results for orientation selectivity and direction selectivity because they are more commonly used in the literature, as the reviewer has noted.

5. The responsiveness measure seems reasonable, but it is based on a seemingly arbitrary threshold of 25% of trials with responses above 95% of the baseline period. What happens if this 25% threshold is varied? In addition, other measures of response reliability might be helpful. For example, trial-to-trial variability might be an interesting and important parameter to quantify, such as using the coefficient of variation or other metrics.

The 25% threshold was chosen based on extensive visual inspection of neural responses. We sought to choose the lowest threshold that reliably detected what appeared to be “true” stimulus-evoked responses, while rejecting likely false positives. The key point for this analysis is that the same threshold is applied to both modalities. Further, we now note in the manuscript that our calculation of J–S distance takes into account the full distribution of reliability, not just the cells above the 25% threshold.

To address the reviewer’s point about trial-to-trial variability, we have included a new Supplementary Figure 7 showing that the coefficient of variation is very closely related to our original response reliability metric.

6. My understanding is that the calcium imaging data were segmented into cells using morphology based methods. This is not the current standard in the field. Most papers now use correlations in the fluorescence timeseries to help with segmentation. Is the segmentation obtained by the authors consistent with the segmentation obtained with more standard methods in the field, such as the cells obtained with Suite2P or CNMF? It might also be helpful for the authors to discuss how cell segmentation with calcium imaging, including neuropil contamination/subtraction, could impact the metrics they studied. The analysis of potential spike sorting artifacts or biases was informative, and something analogous could be, at the least, discussed for calcium imaging segmentation.

The segmentation method we have used is predominantly, though not exclusively, based on morphology. This allows us to identify neurons that are present but only rarely active. It is easy to see how exclusive reliance on activity correlations for cell detection may create a bias towards higher firing rates, similar to the bias present in ephys where units are completely invisible unless they are active. If Suite2P / CNMF suffer from such bias, then using those methods in our comparison would act similar to sub-selecting 2P imaged units by firing rate, which as we have seen, helps to reconcile responsiveness distributions. However, we would be missing out on fully appreciating the extent of bias that would be present in both compared modalities, and actually learn less about the true nature of the biological system in the process.

Moreover, to the best of our knowledge, there is not yet consensus in the field regarding the best segmentation method. Suite2P and CNMF are indeed widely used, but they are by no means the de facto standard. In our own testing, we have found that these two methods produce conflicting segmentation results on most of our experiments, about as different from one another as each would be from our own segmentation results. Additionally, the hyperparameter space for these segmentation methods is fairly high-dimensional and impractical to search exhaustively. Further, all segmentation methods require an annotation step, either manually or through training a classifier, to identify valid ROIs and exclude debris and other artifacts. This annotation step is likely to have further impacts on biases in the population. Overall, we feel that a full and comprehensive sensitivity analysis of how different segmentation methods might affect our results might uncover additional biases but would be beyond the scope of this manuscript.

7. I do not like the introduction of the term 'ophys'. To me, this just adds another term to the field and likely results in more confusion rather than sticking with normal nomenclature in the field. I do not see what it adds over referring to the data as calcium imaging or two photon calcium imaging. If anything 'ophys' seems less precise because it could include voltage imaging too.

We previously used ‘ophys’ as a convenient short-hand to denote our specific dataset as collected via two-photon calcium imaging. To avoid introducing a somewhat arbitrary new term into the literature, we have replaced this with “imaging” throughout the manuscript.

Reviewer #3:This paper makes a step towards addressing a need in the community: understanding if measurements made with electrophysiology (ephys) and 2-photon imaging (2p) give the same result, and if not, in which way they differ and why. The best way to compare the two would be to do ephys and 2p at the same time (e.g. Wei,.… Druckmann, bioRxiv 2019). Next best would be to do them at different times but in the same neurons (which could be done here, because many of the ephys mice express GCaMP6f). This paper uses the third best approach: it does ephys and 2p in different mice. Not ideal but a useful exercise given the remarkably vast datasets.

We agree that characterizing functional metrics in simultaneous 2P and ephys recordings, or serial recordings from the same set of neurons, would alleviate some of the confounds of recording from distinct populations for each modality. However, we would like to point out that such experiments are extremely difficult and low-yield and have not, in fact, been attempted in any publication that we are aware of (including the one by Wei et al.). If one has recorded from a population of cells using twophoton imaging, it would require a heroic effort to sample the exact same set cells using electrophysiology, especially considering the relatively orthogonal geometry of these two modalities. Given current technology, the only practical way to obtain comparable data from large populations of cells is to record each modality separately.

The Wei et al. work (now published in *PLoS Computational Biology*) takes an approach similar to our own: the ephys and 2P recordings they use to compute functional metrics all take place in separate mice. Their simultaneous 2P and ephys recordings are only used to calibrate the spikes-to-calcium forward model. These recordings are typically very short (on the order of a few minutes), and are often performed under anesthetized conditions. We have previously collected a similar set of simultaneous recordings (Ledochowitsch et al., 2019; Huang et al. 2019), which we use for the same purpose. We now include new analysis of this data in Figure 3, in order to demonstrate the impact of event extraction parameters on selectivity and responsiveness metrics.

The paper however contains some flawed analyses and meanders away from the main focus. It could be refocused e.g. by keeping the top half of Figure 1; a revised Figure 2; Figure 4; the top half of Figure 5, and a panel similar to Figure 4E, showing that once biases in cell selection are taken care of, the fraction of responsive neurons to different stimuli becomes the same. The current Figures 2 and 7 show analyses that are simply not ok, as explained below.

We thank the reviewer for these suggestions. Hopefully after reading our responses below, the justification for including all of the original figures will become clear.

(1) The paper starts by making a flawed comparison between 2p and ephys data (Figure 2). One can't directly compare the output of ephys, i.e. spike trains that are sequences of 1s and 0s, with the deconvolved output of 2p, i.e. firing rates that range over 4 orders of magnitude. To compare those two measures of activity they must have the same units: either compute firing rates with common bin size (by binning the spike trains and resampling the deconvolved firing rates) or convolve the spike trains with a filter to match the GCaMP timecourse (as in Figure 4).

The reviewer raises a critical point, which is that an apples-to-apples comparison of 2P and ephys data is tricky, given the fundamental differences between these two modalities. In order to address this shortcoming, we have taken the approach that the reviewer suggests: all the analyses in (original) Figure 4, 5, and 6 are carried out after applying a spikes-to-calcium forward model to the ephys data.

The purpose of Figure 2 is to highlight the differences between 2P and ephys as they are typically analyzed in the literature, prior to any attempt at reconciliation. Besides the Wei et al. paper, such a direct comparison has not been undertaken previously. We therefore feel it is essential to present these “baseline” results before we show the effects of applying the spikes-to-calcium forward model. Even though the underlying signals have different units, we can still derive the same functional metrics for both datasets (e.g., selectivity, responsiveness, and preference). We have chosen to compare these derived metrics, rather than comparing spike trains and 2P events directly. Furthermore, in all cases, we calculate the responses for both modalities using a common bin size (2 s for drifting gratings, 250 ms for static gratings and natural scenes, and 33 ms for natural videos).

We have now updated Figure 2 to include a comparison of the steps involved in extracting response magnitudes from the ephys and imaging data. We have also added a new Figure 3, which shows how the parameters of the DF/F event extraction can bias the functional metrics. We believe it is essential to present this baseline analysis prior to diving into the results obtained after applying the forward model to the ephys data.

(2) Based on Figure 2, the paper finds two discrepancies: the 2p dataset has neurons with higher stimulus selectivity (Figure 2H) but a lower fraction of responsive neurons (Figure 2F). The first discrepancy disappears once the spikes measured with ephys are convolved with a reasonable filter that models the impulse response of GCaMP (Figure 4I). The second discrepancy remains (Figure 4E) but it disappears (does it? this is not fully shown) if one considers that the ephys data is biased towards neurons that fire more.

The reviewer is correct that the two main discrepancies in the baseline analysis (Figure 2) are the apparent higher selectivity in the 2P data, and the apparent higher fraction of responsive neurons in the ephys data. The difference in selectivity can be mostly – but not completely – reconciled by applying the forward model (this is more than just a convolution, as it is highly nonlinear). This is shown in (original) Figure 4I, 6B, 6C, and 6D, and is summarized in Figure 6A. The difference in responsiveness can be mostly

– but not completely – reconciled by filtering out low-event-rate cells from the 2P data (original Figure 5A) or filtering out contaminated cells from the ephys data (Figure 5C). This point was also summarized in the original Figure 6A.

We have stated these conclusions explicitly in the following sentences in the discussion:

“Even after accounting for these known factors to the best of our ability, the overall population from our imaging experiments still displayed higher selectivity than its ephys counterpart.” (p. 23)

“In order to reconcile some of these differences [in responsiveness], we could either apply a very strict threshold on ISI violations score to the ephys dataset, or remove between 61-93% of the least active neurons from the imaging dataset.” (p. 23)

(3) The analysis of cell selection bias takes 3 long figures (Figure 5,6, and 7) without delivering the desired result: we need to see a figure with the same format as Figure 1F and Figure 4I, showing a match between the two techniques – if such a match can be obtained.

We would like to clarify some points for the reviewer:

Only (original) Figure 5 is about cell selection bias. It shows the impact of sub-selecting cells based on their event rate (2P) or contamination level (ephys). Because the curves do not reach 0 (perfectly overlapping distributions), we conclude that sub-selection is not sufficient to fully reconcile the results of both modalities.

Our original Figure 6 summarizes the impact of all of the transformations we have carried out, including layer matching, behavioral state matching, and the spikes-to-calcium forward model, in addition to sub-selecting cells, for all of the stimuli used. We believe that Figure 6A may already contain the information the reviewer was expecting to see. The panels on the right-hand side show the cumulative effects of applying the forward model (yellow shaded region) and the cell selection procedures (“Events” and “ISI” columns). In order to save space, instead of showing the full distribution of selectivities for all areas (as in the original Figure 4I), we just show the J–S distances for each area.

Our original Figure 7 shows how we can apply our knowledge of inter-modal differences to achieve a useful scientific result. The justification for including this figure is explained below.

(4) To address sampling bias introduced by the 2p processing pipeline, the paper explores the effect of setting a threshold at a given firing rate (p 13). However, the sampling bias may be more closely related to the procedures used to identify neurons in the 2p images. It seems that ROIs here are based on morphological properties without further manual curation (de Vries et al., 2020). This wide net will increase the total number of neurons, and thus likely decrease the fraction of responsive neurons, as pointed out by the same group (Mesa et al., bioRxiv). Subsampling using event rate reduces rather than increases overall selectivity, but in the Mesa et al. preprint, subsampling using a higher threshold for coefficient of variation (CV) increases selectivity with fewer neurons. So, at the very least, consider subsampling using CV to see whether this increases similarity between recording methods.

We think it is important to clarify several points here. First, we explored the effect of selecting imaging cells on their event rate in order to examine the sampling bias of electrophysiology, not of the 2P processing pipeline. Second, ROIs do in fact undergo curation using an automated classifier trained on manually annotated data, in order to remove cell fragments or processes from further analysis, as described in de Vries et al. (2020). Third, Mesa et al. (now published in eNeuro) does not claim that simply by recording more neurons one would expect a lower fraction of responsive neurons. Rather, that paper examined the impact of different definitions of “responsiveness” on the resulting population metrics. The authors examined the effects of sub-sampling neurons by *lowering* the coefficient of variation threshold (so only more robustly responding cells are included). When analyzing fewer, more robust neurons, they found that direction selectivity went down, but that orientation selectivity increased slightly for only the *very* most robust neurons. Event rate is (roughly) one axis of CV, but as it doesn’t include any information about variability, it is possible that these could have different effects on the measured metrics. Following the reviewer’s suggestion, we tried sub-sampling the cells in the imaging data based on their coefficient of variation (Supplementary Figure 7). The results appear very similar to the effects of filtering by event rate; cells with lower CV (more robustness) tend to be more responsive, and also have slightly higher selectivity (as measured by lifetime sparseness). But again, this is only for the most robust neurons with the lowest CV. We also include a panel in the same figure showing that CV is strongly negatively correlated with our original metric for response reliability, the fraction of preferred condition trials with event magnitudes higher than 95% of baseline intervals. Based on this, we believe that the responsiveness criterion we used, as a threshold on this response reliability, is effective at identifying robust responses.

(5) Another reason for high sparseness/selectivity in the 2p data could be the deconvolution algorithm. Indeed L0 deconvolution, and L0 regularization in general, are intended to sparsify (e.g. as described in Pachitariu et al., 2018 J Neurosci). This could be dealt with by re-analyzing the dataset with different deconvolution algorithms, or vary the parameter λ governing the L0/L1 trade-off. Or perhaps better, one could make synthetic 2p data by convolving the ephys data and then deconvolve it to estimate firing rates. If they become sparser, there is a problem.

This is a very similar to the main point raised by Reviewer #1, and we believe it is an important one. The choice of λ does have an impact on functional metrics, specifically because it has a very strong impact on the overall sparseness of the deconvolved data (in terms of event rate). As the effect of regularization is lessened, more events are extracted from the 2P data, most of which are false positives that are detected in the absence of a true underlying spike. We originally chose the exact L0 method (and the corresponding λ values) based on its ability to predict firing rates in the ground truth data. But we now realize that there are other features of the event detection step that can affect responsiveness and selectivity metrics.

We have taken several steps to address this issue in the revised manuscript:

1. We have added a new section and main text figure that demonstrate how the approach to deconvolution impacts our results, with an explicit comparison between exact L0 (our original method) and non-negative deconvolution, a method that has been shown to out-perform more complex algorithms on ground truth datasets (p. 8-11)

2. We have added new figure panels 3B-D showing that low-amplitude events can be removed from the data (essentially, performing a post-hoc regularization step) without affecting correlation with ground truth firing rate.

3. We have added figure panels 3E-G showing that we can achieve nearly arbitrary levels of responsiveness and selectivity in the 2P dataset by varying the post-hoc event amplitude threshold. We show that changing the parameters of this thresholding step can reconcile the results from ephys and imaging—however, responsiveness and selectivity are matched at different threshold levels, suggesting that this cannot be the whole store.

We would also like to point out that we do not view the apparent sparsification seen in both real and simulated 2P data as a “problem,” but rather an inherent property of the methodology itself. We know from the ground truth experiments that a significant fraction of spikes do not produce detectable calcium transients (Ledochowitsch et al., 2019). Furthermore, the nonlinear relationship between spike rate and event amplitude (i.e., a few spikes can produce very large events if they occur as part of a burst but smaller events if they are separated in time), leads to further sparsification. The overall goal of our manuscript is to accurately characterize the effect of this sparsification on derived functional metrics, rather than to try to eliminate it.

(6) The statistical test of Jensen-Shannon distance seems to find significant differences also where there barely are any. In some analyses (e.g. Figure 4I) the distributions look highly similar, yet are still judged to be statistically distinct. Is there an instance (e.g. within the same modality) where the Jensen-Shannon distance would ever be small enough to fail to reject the null? For example, by taking two sets of ephys data obtained from the same population in two different days.

Jensen–Shannon distance by itself is not a statistical test. Instead, we apply a bootstrap procedure to determine the distribution of distances that can be expected by sub-sampling the same set of within modality data. For the selectivity differences in Figure 4I, we find that 100% of these bootstrapped values fall under 0.14, the average distance between the inter-modality distributions. Therefore, the distance between the two distributions—although relatively small—is still larger than what would be expected to occur by chance. The details of, and justification for, this procedure are now described more thoroughly in our methods section (p. 31).

(7) Figure 7 shows an unconvincing attempt at clustering neurons. It rests on a clustering process that seems dubious (t-SNE is geared to show data as clusters) and would require a whole paper to be convincing. It hinges on the existence and robustness of clusters of cells based on responses to a variety of stimuli. If there are clusters of that kind, that's a major discovery and it should get its own paper. But showing it convincingly would likely require long recordings (e.g. across days). It is not the kind of thing one shows at the end of a paper on a different topic. What we want at the end of this paper is to know whether the correction for selection bias explains the discrepancy shown in the simple analysis of Figure 4.

The t-SNE plot in (original) Figure 7 is used for visualization purposes only, which we now state in figure caption for (new) Figure 8. The clusters are derived via the Gaussian mixture model approach used in de Vries et al. (2020). The functional clusters we observe are consistent across many repeats of the same procedure with randomization of initial conditions, so we believe they are sufficiently robust to subject to further analysis.

The purpose of including Figure 7 was not to show the distribution of function clusters per se, but rather to highlight how our understanding of baseline differences between 2P and ephys can be applied to a scientific question that goes beyond single-cell functional metrics. We find that a naive application of the same method to the ephys data yields qualitatively different results (Figure 7B). However, once we account for the higher baseline responsiveness in ephys, the distribution of functional clusters is well matched (Figure 7E). We consider Figure 7 to be a satisfying illustration of how the conclusions of the present manuscript can guide future analysis efforts.

If the reviewer still thinks this is too far off topic to include, we would consider instead concluding with Figure 6, which summarizes how well various corrections can reduce discrepancies between the two modalities.

(8) The separate analysis of different visual areas adds complexity without delivering benefits. There is little reason to think that the relationship between spikes and calcium is different in neighboring cortical areas. If it is, then it's an interesting finding, but one that will require substantial burden of proof (as in simultaneous recordings). Otherwise it seems more logical to pool the neurons, greatly simplifying the paper. (This said, it does seem somewhat useful to separate the visual areas in Figure 4G, as they do have some differences in selectivity).

There are a number of reasons why we decided to analyze each area separately:

1. Our ephys and imaging datasets include different relative numbers of cells from each area. For example, the imaging dataset has many more cells from V1. If we aggregated cells across all areas, it would have been necessary to perform sub-selection in order to match the area distributions, which would have decreased the overall sample size. Prior literature has established clear functional differences between mouse visual areas, in terms of their preferred spatial/temporal frequencies (e.g., Andermann et al., 2011; Marshel et al., 2011), and we did not want these underlying functional differences to impact our results.

2. The bursting properties of individual areas differ somewhat (Figure 3D), and we originally thought this could have an impact on the detectability of calcium transients (and hence our ability to detect responsive neurons). We know that overall responsiveness decreases along the visual hierarchy (from V1 to AM; Figure 2F), and this corresponds with a decrease in burstiness (Figure 3D). However, we found that these intrinsic properties did not have an appreciable impact on responsiveness, based on our forward model (Figure 4E). Such conclusions could not be reached without analyzing each area individually.

3. By treating each area as an independent sample (even though some of the cells come from the same mouse), it increases the power of our overall conclusions.

(9) The paper would be much stronger if it included an analysis of GCaMP6s data in addition to GCaMP6f. Are the same findings made with those mice?

At present, we do not have enough data from GCaMP6s mice to say whether or not the results generalize to other types of indicators. However, in our analysis of the forward model parameter sweep (which now appears in Supplementary Figure 6), we see very little effect of fluorescence decay time on functional metrics for the same cells. This indicates that, for the same underlying physiology, we do not see differences when a GCaMP6s-like forward model is used. We have added a paragraph to the discussion that considers this point in more detail (p. 23), which is reproduced here:

“One potential limitation of our approach is that we have only imaged the activity of transgenically expressed calcium indicators, rather than indicators expressed using a viral approach. […] The full range of this sweep includes decay times that are consistent with GCaMP6s, and event amplitudes that are consistent with viral expression.”

(10) Are there consistent differences in reliability of the responses (repeatability across trials of same stimuli) in the two measures, and if so, does the convolution and selection biases explain these differences.

The response reliability is consistently higher in ephys than in imaging, both before and after applying the forward model. In Author response image 1, there is a higher fraction of ephys cells included for every reliability threshold, except at the very lowest levels (the 0.25 threshold we use throughout the manuscript is indicated with a dotted line).

**Author response image 1. sa2fig1:** 

As we show in (original) Figure 4D, the forward model does not change the overall reliability of cells from the ephys dataset. This is because responses to preferred stimuli in ephys often include spike bursts, which are translated into large-amplitude calcium events by the forward model. Therefore, we believe the most likely explanation for the higher reliability seen in ephys is either:*1.* Selection bias for more active cells in ephys – we show in Figure 5A that selecting the most active cells from the imaging dataset can make the response reliability distributions appear more similar.

*2.* Spike train contamination in the ephys data – we show in Figure 5C that selecting the least contaminated cells from the ephys dataset can make the response reliability distributions appear more similar.

[Editors’ note: what follows is the authors’ response to the second round of review.]

Reviewer #1:My earlier review had suggested ways to make this paper briefer and more incisive so that it would have a clearer message. To summarize my opinion, the message would have been clearer if the paper started by listing the differences between the two methods (using an apples-to-apples comparison from the get-go) and then introduced the forward model and step by step indicated what one needs to do to make the differences go away. I would not introduce in the last figure a new result/analysis (the clusters of cell types) and use that one to illustrate the success of the forward model. I would drop that figure because it's a dubious (potentially interesting) result that would need its own paper to be convincing. By the time we arrive at that figure we should have established how we will judge a model's capabilities, rather than throw more data at the reader. This said, there are certainly many ways to write a paper, and it should be ok to ignore one reviewer's opinion on how to arrange an argument.

We thank the reviewer for this constructive feedback. As suggested, we have added an extended new section that explicitly outlines the specific scenarios where inter-modality reconciliation is likely needed, and what steps should be taken to account for apparent discrepancies (“Lessons for future comparison”). As we state in the manuscript, the ultimate goal is not to completely eliminate differences, but to probe the extend to which differences are consistent with known biases inherent to each modality. We aim to call attention to the fact that these biases exist (something that is often overlooked in the literature), and to provide a clearer understanding of their origins. We hope this new section will provide readers with a set of actionable takeaways that they can apply to their own results, as this reviewer suggested.

Second, we shortened, streamlined, and re-framed our presentation of the last figure (Figure 8) in a new section titled “Interpreting higher-order analyses in light of our findings.” We agree with the reviewer that, previously, the purpose of this figure was not sufficiently clear. In this revision, we articulate more clearly and explicitly that this was not meant to be a novel result about neuronal functional classification. Rather, we were revisiting one of our recent published results in order to illustrate a scenario in which applying the same analysis to imaging and ephys data would yield seemingly conflicting conclusions. We show that after accounting for a specific bias (higher responsiveness in the ephys data), our originally published classification result (based on imaging) still remains true. This particular example also provides additional value because it demonstrates that biases affecting responsiveness can have more insidious and non-obvious effects on analysis outcome than merely acting as a filter for neuronal inclusion/exclusion.

My main suggestion is to give the paper another look and try to convey a clearer message. For instance, take this sentence in abstract: "However, the forward model could not reconcile differences in responsiveness without sub-selecting neurons based on event rate or level of signal contamination." If one took out the double negative (not/without) one would end up with a positive result: the forward model can indeed reconcile those differences, provided that one focuses on neurons with low contamination and sets a threshold for event rate. Writing things this way would actually honor the goal stated in the title: "Reconciling functional differences…". Overall, it would be good if readers could come out with a clearer understanding of what they should do to their 2p data to make it more similar to ephys data, and what aspects they can trust and what aspects they cannot.

We fully agree that the manuscript was previously missing an easily digestible summary of our findings. As mentioned above, we have added a new section titled, “Lessons for future comparisons,” addressing this concern.

We also updated the relevant sentence in the abstract, which now reads, “However, the forward model could only reconcile differences in responsiveness when restricted to neurons with low contamination and a threshold for event rate.”